



# Field observations of Volatile Organic Compound (VOC) exchange in red oaks

Luca Cappellin[1,2], Alberto Algarra Alarcon[2,4], Irina Herdlinger-Blatt[3], Franco Biasioli[2], Scot Martin[1], Francesco Loreto[5], Karena McKinney[1]

[1]School of Engineering and Applied Sciences, Harvard University, Cambridge, Massachusetts, USA
[2]Research and Innovation Centre, Fondazione Edmund Mach, S. Michele a/A 38010, Italy
[3]Institute of Ion Physics and Applied Physics, University of Innsbruck, Innsbruck 6020, Austria
[4]Institute of Ecology, University of Innsbruck, Innsbruck 6020, Austria
[5]National Research Council, Department of Biology, Agriculture and Food Science (DISBA), Rome 7-00185, Italy

*Correspondence to*: Luca Cappellin (luca.cappellin@fmach.it) and Karena McKinney (kamckinney@seas.harvard.edu)

## Abstract.

Volatile organic compounds (VOCs) emitted by forests strongly affect the chemical composition of the atmosphere. While the emission of isoprenoids has been largely characterized, forests also exchange many oxygenated VOCs (oVOCs), including methanol, acetone, methyl ethyl ketone (MEK) and acetaldehyde, which are less well understood. We monitored total branch-level exchange of VOCs of a strong isoprene emitter (*Quercus rubra* L.) in a mixed forest in New England, where canopy-level fluxes of VOCs had been previously measured. We report daily exchange of several oVOCs and investigated unknown sources and sinks, finding several novel insights. In particular, we found that emission of MEK is linked to uptake of methyl vinyl ketone (MVK), a product of isoprene oxidation. The link was confirmed by corollary experiments proving *in vivo* detoxification of MVK, which is harmful to plants. Comparison of MEK, MVK and isoprene fluxes provided an indirect indication of within-plant isoprene oxidation.



Furthermore, besides confirming bidirectional exchange of acetaldehyde, we also report for the first time direct evidence of benzaldehyde bidirectional exchange in forest plants. Net emission or deposition of benzaldehyde was found in different periods of measurements, indicating an unknown foliar sink that may influence atmospheric concentrations. Other VOCs, including methanol, acetone and monoterpenes, showed clear daily emission trends, but no deposition. Measured VOC emission and deposition rates were generally consistent with their ecosystem-scale flux measurements at a nearby site.

**Keywords**

Volatile organic compounds, plant emission, plant uptake, atmosphere-biosphere interactions, red oak.



## 1 Introduction

Global budgets indicate that biogenic emissions of volatile organic compounds (VOCs) are about an order of magnitude larger than those from anthropogenic sources (Benkovitz et al., 2004; Guenther et al., 1995). VOCs emitted from forest environments account for approximately half of the reactive carbon introduced globally into the atmosphere (Guenther, 2002), with isoprene alone contributing about a third of the worldwide VOC emissions (Guenther et al., 2006). Biogenic VOCs are typically more reactive than anthropogenic ones and more readily contribute forming secondary organic aerosols (Atkinson and Arey, 1998; Carlton et al., 2009; Claeys et al., 2004; Griffin et al., 1999; Helmig et al., 2006; Kanakidou et al., 2005; Kavouras et al., 1998). Oxygenated VOCs (oVOCs) constitute an abundant class of atmospheric VOCs. oVOCs can be emitted from biogenic and anthropogenic sources, and are often produced by secondary reactions (mainly VOC oxidation) in the atmospheric reactor (Galbally and Kirstine, 2002; Jacob et al., 2002). The most abundant biogenic oVOC in the troposphere is methanol, typically followed by acetone, formaldehyde, and acetaldehyde. A vast number of other oVOCs are also present in the atmosphere (Goldstein and Galbally, 2007). In particular, oxidation products of isoprene, such as methyl vinyl ketone (MVK), methacrolein (MACR), isoprene hydroperoxides (ISOPOOHs), and isoprene epoxydiols, play a critical role in reactive carbon cycling, SOA formation, and modulation of the atmospheric oxidation capacity (Rivera-Rios et al., 2014). Atmospheric oVOCs are removed via SOA formation, photooxidation, or dry and wet deposition. Their atmospheric lifetimes vary widely depending on functionality, from several hours for MVK, MACR, and ISOPOOH, to 10 days for methanol and 1 month for acetone.





Forests are not only major sources of biogenic VOCs but can also act as efficient oVOC sinks. oVOCs can be lost by dry deposition onto plant surfaces or by uptake into the plant through the stomata. Recently, plant uptake has been shown to be a more significant mechanism for oVOC removal than previously assumed (Karl et al., 2010). Plant uptake of oVOCs requires an in-vivo sink, such as enzymatic consumption (Cojocariu et al., 2004). Despite many studies documenting plant uptake, the mechanisms of these sinks, which vary across compounds, are not yet fully understood. Uptake of methanol and acetaldehyde by plants has been reported (Jardine et al., 2008; Laffineur et al., 2012), and mechanisms for their metabolism within plants have been proposed (Gout, 2000; MacDonald and Kimmerer, 1993). As both biogenic sources and sinks of these compounds exist, bidirectional exchange can be observed (Jardine et al., 2008; Karl et al., 2010; Laffineur et al., 2012; Misztal et al., 2011; Schade et al., 2011). Similarly, there are many reports of negative acetone fluxes at ecosystem-level (Karl et al., 2004, 2005), suggesting deposition or plant uptake. Yet, evidence for acetone uptake at plant level is inconclusive (Cojocariu et al., 2004; Tani and Hewitt, 2009), and no mechanism driving acetone uptake by plants is known (Cojocariu et al., 2004).

A plant sink of isoprene oxidation products methacrolein (MACR) and methyl vinyl ketone (MVK) has also been demonstrated (Andreae et al., 2002; Karl et al., 2004, 2005, 2010), suggesting a metabolic consumption of these compounds, especially given that MVK and MACR are quite toxic. For MACR, detoxification mechanisms have already been elucidated (Muramoto et al., 2015), whereas knowledge of metabolic pathways consuming MVK is scant. At the same time, some studies have suggested that isoprene plays an antioxidant role within plants, reacting with reactive oxygen species (ROS) to



produce MVK and MACR, which are then emitted (Fares et al., 2015; Jardine et al., 2012). Production and emission of isoprene oxidation products by plants is still matter of debate, but if it does occur, it could explain why a detoxification mechanism for MVK and MACR also exists. Moreover, very little is known about plant emission and uptake of other common biogenic oVOCs, such as methyl ethyl ketone (MEK). MEK can lead to ozone and PAN production in the atmosphere (Pinho et al., 2005) and photochemical odd-hydrogen formation in the upper troposphere (Atkinson, 2000; Baeza Romero et al., 2005). Recent studies point out the great importance of biogenic MEK sources (Yáñez-Serrano et al., 2016); however, the origin of biogenic MEK remains unclear.

Among observed oVOCs, several belong to the class of benzenoid compounds, which are believed to be synthesized and emitted by vegetation under stress conditions and as chemical signals, although direct observations of such compounds are still very limited (Bouvier-Brown et al., 2009; Heiden et al., 1999; Jardine et al., 2010; Kim et al., 2010; Leone and Seinfeld, 1984; Misztal et al., 2010, 2015; Owen et al., 2002; White et al., 2009). Uptake of benzenoid compounds such as phenol and benzaldehyde has been seen in houseplants but has not been reported in forest environments so far (Kondo et al., 1996; Tani and Hewitt, 2009). In general, the impact of vegetation on the atmospheric concentrations of benzenoid compounds has usually been overlooked. Only recently, the first suggestion emerged that plant emissions of benzenoid compounds might be comparable to those from anthropogenic sources (Misztal et al., 2015).

In general, direct plant-level field observations of the balance between emissions and deposition of oVOCs are scarce. The present study aims to directly investigate exchange





of oVOCs at branch level in a forest environment. The compounds include methanol, acetone, acetaldehyde, MEK, the isoprene oxidation products MACR, MVK, and ISOPOOH, and the aromatic compound benzaldehyde. Isoprene and monoterpenes are also included as references. Emissions and/or uptake of each compound compared with

the literature reports and with recent canopy-scale flux measurements at a nearby site (McKinney et al., 2011). Moreover, observations were corroborated with ancillary fumigation experiments with oVOC standards. The results i) advance the current understanding of oVOC exchange, suggesting plant detoxification of MVK and a mechanistic explanation for the emission of MEK; ii) provide direct evidence of oVOC

exchange at branch level, testing in particular the presence of bidirectional exchange of benzenoids; iii) suggest avenues for further mechanistic studies on oVOC exchange.

## 2 Materials and Methods

## 2.1 Forest site description

Experiments were performed at Harvard Forest, a New England mixed forest located in

Petersham, MA (42.54° N, 72.17° W, elevation 340 m), a rural area in central Massachusetts. Measurements were made from a walk-up tower (30 m) installed in 2013 and surrounded predominantly by red oak trees. The tower is about 200 m SW of the Environmental Measurement Site (EMS) where previous BVOC measurements were taken at ecosystem-level (McKinney et al., 2011; Moody et al., 1998). The tower is

constructed on slightly hilly terrain surrounded by forest for several kilometers. The closest paved road is about 1.5 km and the nearest town is approximately 5 km distant. Influences come from regional-scale transport especially from cities on the East Coast (including Boston, New York, and Albany, located approximately 110 km to the east, 300 km to the





southeast, and 130 km to the west of the forest, respectively). Midday ozone concentrations during summers are typically 40±12 ppbv for periods of NW winds and 58±19 ppbv for periods of SW winds, while NOy (i.e. the sum of nitric oxide, nitrogen dioxide, nitric acid, and organic nitrates) are 2.3±1.4 ppbv and 5.1±2.7 ppbv, respectively (Moody et al., 1998; Munger et al., 1996).

The forest stand is 85-120 y old and has been mainly undisturbed over the past 70 years (Urbanski et al., 2007). The canopy is about 22.5 m high. Measurements of forest leaf area index are routinely made as well as meteorological data (Boose, 2001). Branch enclosure measurements were made using canopy-top branches of red oaks located on the south or east side of the tower.

## 2.2 Vertical BVOC concentration profiles

Air samples were drawn from several heights on the tower (7.5, 15, 22.5, 30 m) through FEP Teflon tubing (0.250 inch outer diameter x 0.156 inch inner diameter) connected to a Teflon valve system (NResearch, Inc.) located at 22.5 m on the tower.  A 40 m FEP Teflon (0.250 in OD x 0.156 in ID) line connected the valve system with a shed located at the bottom of the tower and housing the analytical instrumentation. The air flow rate through the line was set to ca. 11 slm to achieve residence times of ca. 7 s. A filter (2.0-μm pore size 47-mm diameter Zefluor teflon filter, Pall Corp.) was placed at each inlet to prevent particles from entering. Filters were replaced weekly. The valve system was set to randomly cycle between the inlets every 10 min (30 min in the case of tower top inlet). The inlets not sampled during a given measurement cycle were flushed by a second pump. In this way air was flowing continuously though all inlet lines, preventing signal transients



upon valve switching. 200 sccm of the sample flow was drawn continuously by a Proton Transfer Reaction - Time of Flight - Mass Spectrometer (PTR-ToF-MS) equipped with a Switchable Reagent Ion system (PTR/SRI-ToF-MS) for analysis. **Measurements were taken between July 17, 2014, and August 8, 2014.** The instrument was operated in $NO^+$

mode during the period July 17 - 23, and in $H_3O^+$ mode otherwise (See Section 2.4 for details).

Background measurements were made by adding zero air to the inlet line at tower top through a third FEP Teflon line. Zero air was generated by compressing ambient air and delivering it to a catalytic converter consisting of Palladium coated alumina heated to

290°C (Aadco Instruments ZA30). Background measurements were taken automatically every 11 h for 30 min.

## 2.3 Branch enclosure BVOC exchange measurements

Red oak branches accessible from the walk-up tower were selected for branch enclosure

measurements. Three cylindrically shaped PFA Teflon branch enclosures (ca.5 L) were used to encase three unshaded upper canopy branches (22.5 m above ground). At the end of the experiments leaves were collected and leaf areas were measured via the software ImageJ [version 1.47, (Schneider et al., 2012)]. A fourth bag not containing a branch was placed close to the others to serve as a background.

Ambient air was continuously supplied to each enclosure via FEP Teflon tubing (0.250 in OD x 0.156 in ID) at a flow at of ca. 2.5 slm maintained by a pump. Flows were measured by flow meters (Honeywell AWM5000 series) calibrated using Bios DryCal Flow





Calibrators. Branch enclosure air was drawn through FEP Teflon tubing (0.250 in OD x 0.156 in ID) to a Teflon valve system located at 22.5 m on the tower. The valve system previously described for gradient measurements was reconfigured for this purpose. The air flow rate through the sample line was set at 2 slm to achieve residence times of 38 s. Outlet flows were set lower than the inlet flows and excess air was allowed to exit the enclosures through a 30 cm length of FEP Teflon tubing (0.250 in OD x 0.156 in ID). In this way a slight overpressure was created in the enclosures preventing outside air from entering. Each branch enclosure was sampled for 10 min, followed by 10 minutes from the empty enclosure. The valve system was set to randomly cycle among the branch enclosures. The first 3 min of each measurement were discarded to avoid artifacts from the previous sample. The enclosures not sampled during a given measurement cycle were flushed by a second pump. In this way air was flowing continuously though all enclosures and inlet lines, preventing signal transients upon valve switching.

200 sccm of the sample flow was drawn continuously by a PTR/SRI-ToF-MS instrument for analysis. VOC exchange rates were computed from differences between the concentration in the branch enclosure and those in the empty bag, converted in nmol m$^{-2}$ h$^{-1}$ using the measured leaf area and ingoing flow rate. Measurements were taken between August 14, 2014, and September 1, 2014. The instrument was operated in NO$^+$ mode during the periods August 14 - 21 and August 28 - September 1, and in H$_3$O$^+$ mode otherwise (See Section 2.4 for details).

On September 2 ancillary fumigation experiments were carried out. Gas cylinders containing known amounts of the target VOCs (Scott Specialty Gases, Inc.) were diluted



with zero air using flow controllers (MKS Instruments) and delivered to the branch enclosures. Flow rates were set as previously described. For each concentration step, the enclosures, including the empty one, were measured in random order and the signal was allowed to stabilize before a measurement was made. VOC exchange rates were computed as described above.

## 2.4 PTR/SRI-ToF-MS operation and data analysis

VOC measurements were performed by a PTR/SRI-ToF-MS 8000 (Ionicon Analytik GmbH, Innsbruck Austria) equipped with a switchable reagent ion system (Jordan et al., 2009), allowing either $NO^+$ or $H_3O^+$ primary ion chemistry. In $H_3O^+$ mode the drift tube conditions were: 2.19 mbar drift pressure, 542 V drift voltage, 60 °C drift tube temperature. The resulting *E/N* ratio was ca. 125 Townsend (Td) (*N* corresponding to the gas number density and *E* to the electric field strength; 1 Td=$10^{-17}$ $Vcm^2$). The PTR-ToF-MS ion source produces $H_3O^+$ primary ions at high purity, with a fraction of the spurious primary ions $NO_+$ and $O_2^+$ of 0.04-0.3% and 0.8-1.2%, respectively, with respect to the $H_3O^+$ ion signal. In $NO^+$ mode the drift tube conditions were: 2.21 mbar drift pressure, 296 V drift voltage, 90 °C drift tube temperature. The resulting *E/N* ratio was ca. 74 Townsend (Td). During the measurements the fraction of the spurious primary ions $NO_2^+$, $O_2^+$ and $H_3O^+$ were 1.9-2.3 %, 0.1-0.5 % and 0.1-0.3 % respectively, relative to the $NO^+$ signal. A time-of-fight (ToF) mass analyzer operated in its standard configuration (V mode) was used to separate and detect the ions exiting the drift region. The sampling time per channel of ToF acquisition was 0.1 ns, amounting to 350,000 channels for a m/z spectrum ranging up to m/z = 400.



### 2.4.1 NO$^+$ chemistry in SRI-ToF-MS instrument

Previous ion chemistry investigations (Amelynck et al., 2005; Jordan et al., 2009; Knighton et al., 2009; Spanel and Smith, 1998) employing NO$^+$ as primary ion have shown that four reaction pathways dominate. They feature charge transfer,

$$NO^+ + RH \xrightarrow{k_1} RH^+ + NO,$$

(5a)

hydride ion transfer,

$$NO^+ + RH \xrightarrow{k_2} R^+ + HNO,$$

(5b)

hydroxide ion transfer,

$$NO^+ + ROH \xrightarrow{k_3} R^+ + HNO_2,$$

(5c)

and three body association reactions,

$$NO^+ + R + N_2 \xrightarrow{k_4} (NO^+) \cdot R + N_2,$$

(5d)

Isoprene and monoterpenes have ionization potentials lower than NO and charge transfer reactions with NO$^+$ ions (5a) occur at collision rates (Karl et al., 2012). Reaction (5a) does not proceed for VOCs having ionization potentials higher than that of NO (9.26 eV) (Ebata

et al., 1983).

The reaction of aldehydes with NO$^+$ typically proceeds mainly through hydride transfer (channel 5b), whereas that of ketones is mainly through three body association (channel





5d). As a result, in contrast to $H_3O+$, $NO^+$ chemistry allows separation of isomeric ketones and aldehydes (Jordan et al., 2009). For example, in $H_3O+$ mode, MACR and MVK ($C_4H_6O$) are indistinguishable as both are detected in the protonated ion form at m/z 71.049 ($C_4H_7O^+$). In $NO^+$ mode MACR yields m/z 69.033 ($C4H5O^+$) from reaction 5b and, at a

much reduced rate, m/z 100.039 ($C4H6O \cdot NO^+$) from reaction 5. MVK yields only m/z 100.039 from reaction 5d (Liu et al., 2013). Thus, concentrations of MVK and MACR can be independently determined (Liu et al., 2013). Acetone and MEK also proceed via three body association (5d) and yield ions at m/z 88.039 ($C_3H_6O \cdot NO^+$) and m/z 102.055 ($C_4H_8O \cdot NO^+$), respectively. Aldehydes such as acetaldehydes and benzaldehyde react at

collision rate via mainly hydride ion transfer (5b). Methanol does not react with $NO^+$.

### 2.4.2 $H_3O^+$ chemistry

The traditional operational mode of PTR/SRI-ToF-MS is $H_3O^+$ ion chemistry, indicated as PTR-ToF-MS. We refer to the reviews on the subject for details (Blake et al., 2009; de Gouw and Warneke, 2007). Briefly, the analyte gas mixture enters the drift tube, where

molecules having a Gibbs energy of protonation (typically approximated with proton affinity) higher than water by more than 35 kJ·mol[-1] (Bouchoux et al., 1996) efficiently react with the $H_3O^+$ primary ions by proton transfer

$$H_3O^+ + A \xrightarrow{k_5} AH^+ + H_2O.$$

(6)

Alkenes and oVOCs, including all of those in this study, typically exhibit proton affinities high enough for detection using $H_3O^+$ ion chemistry.



### 2.4.3 Spectral analysis

VOC concentrations were calculated from the ion signal at each corresponding mass-to-charge ratio. To correct for count losses due to the detector dead time, an off-line procedure based on Poisson statistics was applied (Cappellin et al., 2011a; Titzmann et al., 2010). An internal mass calibration procedure was used in order to achieve a mass accuracy better than 0.001 Th (Cappellin et al., 2010). Further processing included baseline removal and peak area extraction using the procedure described by (Cappellin et al., 2011b). An optimized peak shape determined from the measured spectra was used for fitting.

The instrument response to each VOC was calibrated using standard gas cylinders (Scott Specialty Gases/Air Liquide). Gas standard 1 contained isoprene (80.0 ppm). Gas standard 2 contained isoprene (1.07 ppm) and MACR (1.07 ppm). Gas standard 3 contained MEK (1.09 ppm), MVK (1.09 ppm), acetone (1.09 ppm), acetaldehyde (1.09 ppm), and methanol (1.09 ppm). Zero air generated as described previously was used for dilution. In the case of compounds for which gas standards were not available (benzaldehyde, 2-butanol, 3-buten-2-ol, monoterpenes), absolute concentrations were estimated with a theoretical approach (Cappellin et al., 2012) using reaction rates with the primary ion computed at the set drift tube conditions (Su, 1994). This method was shown to estimate concentrations within a 10% uncertainty, under certain instrumental conditions (Cappellin et al., 2012; Müller et al., 2014).

Isoprene signal was monitored on the ion peaks 69.070 ($C_5H_9^+$) for $H_3O^+$ mode and 68.062 ($C_5H_8^+$) for $NO^+$ mode (Karl et al., 2012). At the site interferences from MBO in $H_3O^+$ mode are negligible, given the species emitting the BVOCs (Harley et al., 1998; McKinney et al., 2011). Methanol is detected at m/z 33.033 in $H_3O^+$ mode. Previous studies have



established that it is free from contaminations (de Gouw et al., 2003; Warneke et al., 2003). In $NO^+$ mode it was not possible to detect methanol as it does not react with the primary ion. MEK was monitored on the ion peaks 73.065 ($C_4H_9O^+$) for $H_3O^+$ mode and 102.055 ($C_4H_8O \cdot NO^+$) for $NO^+$ mode. Influence of butanal in $H_3O^+$ mode might be present.

However, no butanal has been found to be emitted by red oak branches (Helmig et al., 1999). Moreover, it has not been detected in past investigations using GC separation prior to PTR-MS analysis (de Gouw et al., 2003). Acetone was monitored on the ion peaks 59.049 ($C_3H_7O^+$) for $H_3O^+$ mode and 88.039 ($C_3H_6O \cdot NO^+$) for $NO^+$ mode. Influence from propanal in $H_3O^+$ mode, although possible, is unlikely at the site (de Gouw et al., 2003;

McKinney et al., 2011; Warneke et al., 2003), as it is a compound usually present in urban areas. Acetaldehyde is detected at m/z 45.033 ($C_2H_5O^+$) in $H_3O^+$ mode. Acetaldehyde is not reported in $NO^+$ mode as the main ion 43.018 ($C_2H_3O^+$) is a common fragment, therefore not a reliable signal for acetaldehyde. The ion at m/z 74.024 ($C_2H_4O \cdot NO^+$), from the association reaction of acetaldehyde with $NO^+$ had a weak signal and was discarded.

Benzaldehyde was monitored on the ion peaks 107.049 ($C_7H_7O^+$) for $H_3O^+$ mode and 105.033 ($C_7H_5O^+$) for $NO^+$ mode. Given the relatively distinctive molecular formula of the aromatic benzaldehyde and the very good consistency between $H_3O^+$ and $NO^+$ data, influences on the signal from other compounds are not expected. Monoterpenes [generally constituted by numerous compounds sharing the same m/z, (Loreto et al., 1996)] were

monitored on the ion peaks 137.132 ($C_{10}H_{17}^+$) for $H_3O^+$ mode and 136.125 ($C_{10}H_{16}^+$) for $NO^+$ mode. (de Gouw and Warneke, 2007) show that monoterpene measurements are free from interferences in $H_3O^+$ mode. No contaminants are expected in $NO^+$ mode.





In $H_3O^+$ mode MACR and MVK lead to indistinguishable isomeric ions at m/z 71.049 ($C_4H_7O^+$). Moreover, isoprene hydroxy hydroperoxide isomers (ISOPOOH) decompose on instrument surfaces to form MVK or MACR, which is also detected at m/z 71.049 (Liu, 2015; Liu et al., 2013; Rivera-Rios et al., 2014). As discussed in Section 2.4.1, in $NO^+$

mode MACR is detected mainly at m/z 69.034 ($C_4H_5O^+$) with a small contribution at m/z 100.039 ($C_4H_5O\cdot NO^+$) whereas MVK is detected only at m/z 100.039. The isoprene hydroperoxide isomers 4,3-ISOPOOH (ISOPDOOH) and 3,4-ISOPOOH (ISOPCOOH) decompose to MACR and so can also produce a signal at m/z 69.034 (Liu et al., 2016; Rivera-Rios et al., 2014). ISOPCOOH concentrations are, however, not relevant under

atmospheric conditions and can be neglected. Therefore this peak corresponds to MACR + ISOPDOOH. Analogously, in $NO^+$ mode 1,2-ISOPOOH (ISOPBOOH) produces MVK which yields product ions at m/z 100.039 (Liu et al., 2016; Rivera-Rios et al., 2014). After correction for the contribution from MACR + ISOPDOOH, the peak at 100.039 thus corresponds to the sum MVK + ISOPBOOH.

In $H_3O^+$ mode alcohols commonly undergo proton transfer reactions often followed by the elimination of a water molecule, while in $NO^+$ mode alcohols typically react via hydride ion transfer (Spanel and Smith, 1997). In the ancillary fumigation experiments the alcohols 2-butanol and 3-buten-2-ol were monitored on 73.065 ($C_4H_9O^+$) and m/z 71.049 ($C_4H_7O^+$),

respectively, in $NO^+$ mode. In $H_3O^+$ mode they were monitored on m/z 57.070 ($C_4H_9^+$) and 55.054 ($C_4H_7^+$), respectively (Spanel and Smith, 1997).



## 3 Results and Discussion

Averaged rates of exchange of the most important oVOCs and volatile isoprenoids measured in the present experiments at Harvard Forest are reported in Table 1 and 2 and Figure 1. The results differ strongly among compounds, and each compound is discussed separately below. The main focus is investigating factors affecting bidirectional exchange of oVOCs with the forest upper canopy. Vertical concentration profiles and corollary fumigation experiments are used to corroborate conclusions.

### 3.1 Isoprene

As expected, isoprene emission showed a diel cycle peaking in the middle of the day and dropping to zero at night (Figure 1). Isoprene emission during daytime averaged to ca. $3 \cdot 10^4$ nmol m$^{-2}$ h$^{-1}$ (ca. 2 mg m$^{-2}$ h$^{-1}$) on a leaf area basis, with peaks of $32 \cdot 10^4$ nmol m$^{-2}$ h$^{-1}$ (22 mg m$^{-2}$ h$^{-1}$). No isoprene uptake was detected. Likewise, ancillary experiments (Table 1) showed no deposition during isoprene fumigation of the same upper canopy branches. Vertical isoprene concentration profiles (Figure 2) are also consistent with a source within the canopy during the day and no emissions or deposition at night.

Literature findings are in agreement with these results. Sharkey et al. (1996), reported emissions of isoprene of ca. $15 \cdot 10^4$ nmol m$^{-2}$ h$^{-1}$ (at 30°C and PPFD of 1000 µmol m$^{-2}$ s$^{-1}$) from red oak leaves in a mixed forest. McKinney et al. (2011) reported isoprene canopy fluxes at an adjacent site in the range ca. 1-8 mg m$^{-2}$ h$^{-1}$ on a ground area basis. This magnitude is also consistent with the current results, which correspond to daytime average isoprene fluxes of ca. 2-3 mg m$^{-2}$ h$^{-1}$ on a ground area basis. The branch enclosure data was scaled to ecosystem-level fluxes considering that the average LAI was 5.12 and



assuming that the only isoprene emitter in the forest footprint is red oak, representing c.a. 41% of total leaf area where the fluxes were measured, and that the reduction in isoprene emission due to inhomogeneity throughout the canopy is 25-50%. To the best of our knowledge, no isoprene uptake has been reported to date.

## 3.2 Isoprene oxidation products

The major atmospheric source of MVK, MACR, and ISOPOOH is gas-phase isoprene oxidation. If there is a plant or surface sink of these compounds, a negative flux would be expected. The existence of a sink is supported by many canopy-level studies, which show

deposition of MVK + MACR[1] to forest canopies (Andreae et al., 2002; Karl et al., 2004, 2005, 2010), as well as by plant-level studies that suggest both surface deposition and plant uptake of these compounds may occur (Fares et al., 2015). On the other hand, episodes of positive canopy fluxes of MACR + MVK have been detected at some sites (Spirig et al., 2005). These could be due to spatial gradients and inhomogeneities in isoprene oxidation

or possibly to a biogenic source. To date, emission of isoprene oxidation products directly from plants has been hypothesized but rarely reported (Jardine et al., 2012). A plant source of these compounds could be masked at the canopy scale if there is also a plant sink.

Branch enclosure measurements of isoprene oxidation product fluxes from high isoprene

emitters such as red oak are challenging due to possible interferences from the high levels of isoprene present in the enclosures during daylight. These interferences can arise from gas phase oxidation of isoprene in the sampling system or in the PTR-MS drift tube. As a

---

[1] Previous studies reporting fluxes of MVK+MACR were actually measuring MVK+MACR+ISOPOOH (Rivera-Rios et al., 2014).





result, only measurements during darkness are reported for isoprene oxidation products. For daytime, a constraint of 38 nmol m$^{-2}$ h$^{-1}$ could be inferred for MVK+MACR+ISOPOOH maximum emission but the actual presence of emissions could not be proved. During nighttime, no net emission of MACR+MVK+ISOPOOH was

detected. Deposition of MACR+MVK+ISOPOOH (Figure 1) up to ca. -23 nmol m$^{-2}$ h$^{-1}$ (Table 1) was measured during darkness on August 25-28, when ambient air concentrations of MACR+MVK+ISOPOOH were higher than usual (up to ca. 1 ppbv). Karl et al. (2010) suggested that the uptake of MVK and MACR by vegetation occurs through stomata and is described by Fick's law. The presence of a sustained negative flux

suggests plant uptake, which implies that stomata were still (partially) open at night. This is possible (Niinemets and Reichstein, 2003), and corroborated by the fact that the water vapor emission signal (m/z 48.008 in NO$^+$ mode, corresponding to H$_2$O·NO$^+$) remains positive at night, although decreasing with respect to the daytime emission (data not shown). More insight was provided by the ancillary experiments (Table 3 and Figure 3)

where an inverse relationship between deposition rate and MACR (or MVK) concentration is established during either day or night, consistent with Fick's law (Karl et al., 2010).

The absence of uptake saturation (Figure 3) suggests the existence of a degradation mechanism for isoprene oxidation products inside leaves. Isoprene oxidation products are

thought to be cytotoxic and their rapid removal may increase plant survival under stress conditions (Oikawa and Lerdau, 2013; Vollenweider et al., 2000). Muramoto et al. (2015) studied the uptake of MACR by tomato plants and proved that MACR concentrations up to more than 100 ppmv are efficiently degraded by glutathionylation and enzymatic reduction within the leaves. Glutathionylation may be a more efficient mechanism for





MACR removal than other hypothesized pathways, e.g. enzymatic degradation by aldehyde dehydrogenase (Karl et al., 2010; Kirch et al., 2004). Muramoto et al. (2015) identified release of isobutyraldehyde (5.8% of the fumigated MACR) produced by reduction of MACR. Reduction of carbonyl compounds can therefore be a mechanism of detoxification of oVOC deposited to vegetation.

Contrary to MACR, the fate of MVK taken up by leaves is unknown. Interestingly, in fumigation experiments, the observed uptake of MVK was matched by an emission of MEK, 2-butanol and 3-buten-2-ol of similar magnitude in total (see e.g. Fig. 3). Specifically, emission of MEK accounted for 82±5% of MVK uptake, while emission of 2-butanol and 3-buten-2-ol accounted for 14±4% and 4±1%, respectively. This finding suggests complete conversions of MVK into other volatiles, predominantly MEK. A consistent explanation is enzymatic reduction. The reduction of the alkene moiety of MVK leads to MEK formation. It is hypothesized that this reduction may be catalyzed by MEK oxidoreductase, an NAD(p)-dependent enone reductase. Enzymes with enone reductase activity have been found in prokaryotes and eukaryotes, including plants (Kergomard et al., 1988; Shimoda et al., 2004; Tang and Suga, 1994; Wanner and Tressl, 1998). For instance Wanner and Tressl (1998) purified and characterized two enone reductases from the cytosolic fraction of *Saccharomyces cerevisiae*, and show that they accept MVK as a substrate to form MEK. Shimoda et al. (2004) showed that cell extracts of cyanobacterium Synechococcus sp. PCC7942 reduced a variety of cyclic and acyclic enone substrates, including MVK, to their corresponding alkyl ketones. Enone reductases in other organisms can also catalyze this transformation. Kergomard et al. (1988) demonstrated that unsaturated ketones were reduced by various plant cells. The formation of 3-buten-2-ol is



obtained via reduction of the carbonyl moiety of MVK and 2-butanol is the result of the reduction of both the carbonyl and the alkene moiety (French, 1970). The ratio of 3-buten-2-ol and MEK production, namely 0.05±0.02, provides an indication of the relative importance of the two pathways (reduction of the carbonyl moiety first or reduction of the

alkene moiety first) for MVK detoxification.

## 3.3 Methyl ethyl ketone (MEK)

Branch level emission of MEK in red oak upper canopy ranged between 0-460 nmol m$^{-2}$ h$^{-1}$ (Table 1, Table 2). Emissions peaked in the middle of the day and dropped at night. Daytime emissions averaged 40 nmol m$^{-2}$ h$^{-1}$, decreasing to 5 nmol m$^{-2}$ h$^{-1}$ at nighttime.

When scaling from branch level to canopy-scale under the same assumptions used in Section 3.1, daytime fluxes in the range 3-12 µg m$^{-2}$ h$^{-1}$ on a ground area basis can be estimated. No deposition of MEK was measured at branch level during the campaign.

Very little is known about MEK emission and deposition in forest environments. Kim et

al. (2010) detected significant amounts of MEK over a Ponderosa pine forest in Western Colorado during both days and nights in summer, and Karl et al. (2005) measured positive fluxes of MEK over a loblolly pine plantation. In laboratory experiments, Holzinger et al. (2000) measured emission of MEK from leaves of young *Quercus ilex* trees in the range of 5-13 nmol m$^{-2}$ min$^{-1}$. MEK was also detected in fire plumes (de Gouw et al., 2006), over

pasture (Kirstine et al., 1998) and over cut grass and clover (de Gouw et al., 1999). During summer 2005 and 2007, McKinney et al. (2011) measured at Harvard Forest ecosystem-scale fluxes of m/z 73, which was attributed to MEK, with daytime emissions of 20-80 µg m$^{-2}$ h$^{-1}$ on a ground area basis, comparable to those found in our experiments.



McKinney et al. (2011) reported slightly negative MEK fluxes during nighttime, which were not seen in our experiment on branches. Perhaps at night MEK sinks other than leaf uptake are present. Our nighttime vertical profiles of MEK (Fig. 2) show a decreasing

trend with height, with MEK concentrations lower under the canopy (0.12 ppbv on average) than within (0.18 ppbv) and above canopy (0.2 ppbv). This may be consistent with a ground sink causing an ecosystem-scale negative flux of MEK during nighttime. Karl et al. (2005), measuring vertical MEK profiles within a loblolly pine plantation, also deduced a ground sink. MEK uptake in houseplants (Tani and Hewitt, 2009) and in

*Camellia sasanqua* (Omasa et al., 2002) has been reported, although the authors noted limitations in the metabolic capacity for MEK uptake by leaf tissues (Omasa et al., 2002). In contrast, *Populus nigra* does not take up MEK (Omasa et al., 2000).

The metabolic pathway leading to the emission of MEK by plants is still unclear. As

discussed in Section 3.2, MEK might be produced by MVK detoxification via enzymatic reduction. During daytime, the sources of MVK that might undergo reduction in the leaf are twofold. First, there is uptake of atmospheric MVK. Second, given the fact that *Quercus rubra* is a strong isoprene emitter, MVK might be produced inside the leaves upon reaction of reactive oxygen species (ROS) with isoprene. ROS produced upon heat

stress in genetically engineered *Arabidopsis* and tobacco plants has been shown to react with isoprene (Velikova et al., 2008; Vickers et al., 2009) and this is thought to produce cytotoxic MVK, which may eventually be released in the atmosphere (Fares et al., 2015; Jardine et al., 2012). We have shown that MVK is detoxified into MEK, whose emission is therefore an indicator of MVK presence in leaves. As an indirect confirmation for this





argument, during nighttime, when no isoprene is produced (Loreto and Sharkey, 1990),
isoprene oxidation does not occur, and MEK is generally not formed. Moreover, as already
mentioned, nighttime deposition of MVK was occasionally observed (e.g. on 25-28
August), which should trigger MEK emission. As expected, during nighttime a significant

negative correlation was found (Fig. 4; Pearson correlation coefficient 0.75; p<0.001), with
MEK emission rates not exceeding the corresponding deposition rates of
MVK+MACR+ISOPOOH, which represent the upper limits for MVK uptake rates.

Due to unreliable estimates of daytime MVK uptake, a plot with daytime MEK emission

(similar to Fig. 4) cannot be produced. However, maximum MVK uptake rates can be
estimated from measured MVK concentrations based on the ancillary MVK fumigation
experiments. For example, maximum MVK uptake rates of ca. 4 nmol m$^{-2}$ h$^{-1}$ can be
calculated for the period 21-25 August. However, the emission rates of MEK reached
about 40 nmol m$^{-2}$ h$^{-1}$ during this same period, suggesting a source of MEK significantly

larger than the predicted uptake and reduction of atmospheric MVK. There may be
different explanations for this observation. Perhaps MEK is formed by sources other than
MVK or there exists a within-plant source of MVK which is further detoxified to MEK
and other minor products. Within-leaf oxidation of isoprene to MVK by ROS (Jardine et
al., 2012) could be such a source. Interestingly, we observed a very high correlation

between isoprene and MEK emissions during light hours (Pearson correlation coefficient
in the range 0.75-0.96 for each study period; p<0.001), which may indicate a link between
MEK and isoprene formation leaves. Furthermore, during $H_3O^+$ acquisition mode the
spectral peaks at m/z 57.070 and m/z 55.054, attributed to 2-butanol and 3-buten-2-ol,
respectively, displayed a significant correlation with MEK diel emission profile (Pearson



correlation coefficients 0.97, p<0.001; and 0.82, p<0.001, respectively). Interference of other alcohols at m/z 57.070 and m/z 55.054 is possible, however, GC-MS analysis of red oak volatiles did not report any potentially interfering alcohols (Helmig et al., 1999). The ratio between 3-buten-2-ol emission and MEK emission was 0.07±0.05 (mean ratio of total daytime emissions ± standard deviation), comparable with the corresponding ratio found in MVK fumigation experiments (0.05±0.02), indicating the relative importance of the two enzymatic reduction pathways for MVK detoxification.

## 3.4 Benzaldehyde

Both emission and deposition of benzaldehyde by upper canopy branches was detected (Figure 1). On average the net daily benzaldehyde exchange was positive in the period 14-25 August, with a mean daily flux of 0.4 nmol $m^{-2}$ $h^{-1}$, and negative on 25 August - 1 September, with a mean daily deposition rate of -1 nmol $m^{-2}$ $h^{-1}$ (Table 2). Emissions of up to 2.2 nmol $m^{-2}$ $h^{-1}$ or 0.23 µg $m^{-2}$ $h^{-1}$ were measured, with a peak at midday (Table 1, Figures 1 and 5). In the last two measurement periods, however, benzaldehyde was mainly deposited, at rates up to -7 nmol $m^{-2}$ $h^{-1}$ or -0.7 µg $m^{-2}$ $h^{-1}$ (Table 2), and again with a clear diel cycle. Deposition rates dropped at night but were still significant. Vertical concentration profiles in July (Figure 2) are consistent with a canopy sink and, probably, a ground source during daytime, while nighttime profiles might indicate a canopy/ground sink.

Anthropogenic sources, such as evaporation or combustion of fossil fuels, typically have been considered the main benzenoid sources in the atmosphere (Rasmussen and Khalil, 1983). It is also formed in the atmosphere through the photochemical degradation of





toluene (Atkinson et al., 1980; Leone and Seinfeld, 1984) or from other precursors such as styrene and methylstyrene (Grosjean, 1985). Misztal et al. (2015) recently estimated that biogenic emissions of benzenoid compounds including benzaldehyde rival these other sources.

Benzaldehyde is an aromatic benzenoid compound naturally produced as a plant volatile (Graedel, 1978) and component of floral scents (Knudsen et al., 2006; Pichersky et al., 2006). Benzaldehyde in plants is generally synthetized from phenylalanine (Boatright et al., 2004; Dudareva et al., 2006), which is produced via the shikimate pathway (Herrmann, 1995). Emissions of benzaldehyde have been reported from *Populus tremula* L. x *tremuloides* Michx [0.05 µg m$^{-2}$ h$^{-1}$, (Misztal et al., 2015)], wheat (Batten et al., 1995), creosotebush (Jardine et al., 2010), and previously from red oaks [ca. 0.1 µgC gdw$^{-1}$ h$^{-1}$, (Helmig et al., 1999)]. Emission from plants is usually associated with stress conditions. Benzalzehyde emissions have been measured upon both abiotic (heat) and biotic (herbivore feeding) stresses (Misztal et al., 2015).

A linear relationship between benzaldehyde deposition flux and benzaldehyde concentration (Fig. 6) indicates benzaldehyde exchange is regulated according to Fick's law (Karl et al., 2010; Omasa et al., 2002). Uptake of benzaldehyde by stomata has been suggested in houseplants (Tani and Hewitt, 2009). Daytime uptake rates of benzaldehyde suggest the existence of a compensation point (Figure 6), of around 0.02-0.025 ppbv. Further studies in controlled conditions would be needed to confirm the presence of a compensation point and to elucidate its dependence on environmental and physiological parameters. Further investigations are also required to explain why benzaldehyde is

deposited. Early studies demonstrated that benzenoid compounds such as benzene and toluene may be assimilated by crop plants grown under sterile conditions via aromatic ring cleavage and successive incorporation of their carbon atoms into nonvolatile organic acids or amino acids (Durmishidze, 1975; Ugrekhelidze et al., 1997; Ugrekhelidze, 1976).

Figure 6 also shows that benzaldehyde emissions were detected during the first measurement period when deposition would be expected on the basis of its gas-phase mixing ratio. As mentioned, benzaldehyde emissions from leaves are typically associated with stress conditions (Misztal et al., 2015). This might have been the case, even though it was not possible to determine if a stress was the driving factor.

## 3.5 Acetaldehyde

Bidirectional exchange of acetaldehyde was observed (Fig. 1, Table 1, Table 2), with the strongest emissions (up to 170 nmol $m^{-2}$ $h^{-1}$ on a leaf area basis) occurring in the middle of the day. Significant acetaldehyde deposition occurred mostly in the afternoon in the period 25-28 August, with uptake rates up to -40 nmol $m^{-2}$ $h^{-1}$. Vertical acetaldehyde profiles from the period 25 July - 8 August show a decrease in nighttime acetaldehyde concentrations going from above canopy to below canopy (Fig. 2), suggesting a forest sink of acetaldehyde during nighttime, although the nature of the sink deserves further investigation.

Acetaldehyde uptake by plant canopies from the atmosphere occurs primarily via stomata (Karl et al., 2005; Rottenberger et al., 2004) whereas surface deposition is minor (Jardine





et al., 2008). Kondo et al. (1998) demonstrated that a biological sink for acetaldehyde should exist within plants since they are capable of continuous uptake for as long as 8h even under ambient acetaldehyde concentrations exceeding 200 ppbv.

Jardine et al. (2008) demonstrated that acetaldehyde exchange rates are controlled by ambient acetaldehyde concentrations, stomata resistance to acetaldehyde, and acetaldehyde compensation point, which suggests a plant source as well as a sink. The cause of acetaldehyde emissions from plants remains elusive. Acetaldehyde might be formed in leaves from ethanol translocated from roots, especially when anoxic conditions
prevent root respiration (Kreuzwieser et al., 1999) or from cytosolic pyruvate, e.g. after darkening (Karl et al., 2002a) or wounding (Loreto et al., 2006). In the period of study, we report net acetaldehyde emission from the upper canopy branches, which may be due to any of these sources.

The measured bidirectional flux of acetaldehyde implies a compensation point in the range of 0-3 ppbv. During transitions from emission to uptake (e.g. on 25-28 August) the value was better constrained to 1-2 ppbv. The acetaldehyde compensation point seems to increase with increasing light intensity as might be deduced comparing light intensity, acetaldehyde mixing ratios and fluxes (see e.g. Fig. 1, 25-28 August). This might imply a
light control on acetaldehyde exchange in the field, in line with the findings of Jardine et al. (2008) in laboratory experiments.





## 3.6 Acetone

A great variety of plants have constitutive emissions of acetone (Bracho-Nunez et al., 2013; Isidorov et al., 1985). High acetone concentrations have been reported above several forests [e.g. (Geron et al., 2002; Helmig et al., 1998; Karl et al., 2003; Müller et al., 2006; Pöschl et al., 2001)], tree plantations (Brilli et al. 2014a) and after forest fires (Brilli et al. 2014b). Red oaks also emit acetone (Steinbrecher et al., 2009). In our branch-scale measurements acetone emission reached 340 nmol m$^{-2}$ h$^{-1}$ on a leaf area basis (Table 1). A diel cycle was found with average daytime emissions of 70 nmol m$^{-2}$ h$^{-1}$. Emission dropped at night to an average of 15 nmol m$^{-2}$ h$^{-1}$. Similar diel patterns of acetone emissions, i.e. displaying mid-day emission maxima, have been reported in several canopy-scale experiments (Goldstein and Schade, 2000; Karl et al., 2003, 2002b; Rinne et al., 2007; Schade and Goldstein, 2001).

The branch-scale measurements can be scaled up to daytime average canopy-scale fluxes of about ca. 20-40 μg m$^{-2}$ h$^{-1}$ on a ground area basis. In this case, the assumptions for scaling the data to ecosystem-level fluxes were that red oak leaves emit acetone homogenously throughout the canopy; that average LAI was = 5.2; and that all tree species in the forest emit acetone at the same rate. These values are consistent with the measurements of McKinney et al. (2011) at Harvard Forest, who observed average daytime canopy-scale acetone fluxes between 20-40 μg m$^{-2}$ h$^{-1}$ on a ground area basis in 2007 and a factor 5 higher in 2005.

Dependence of biogenic acetone emission on temperature has been consistently reported (Bracho-Nunez et al., 2013; Cojocariu et al., 2004; Grabmer et al., 2006; Janson and de





Serves, 2001; Schade and Goldstein, 2001), whereas the relationships with light and stomata conductance are less clear (Bracho-Nunez et al., 2013; Cojocariu et al., 2004; Filella et al., 2009). On the other hand, relative humidity is believed to have a negative influence on acetone canopy fluxes (McKinney et al., 2011). Indications in the same direction have been found by Cojocariu et al. (2004) studying acetone emissions from *Picea abies*. The metabolic mechanisms leading to acetone release have not yet been proved. At present, no process-based model is capable of reliably describing biogenic acetone emissions. Selected plants decompose cyanogenic glucosides to generate acetone cyanohydrin, which is metabolized by a lyase to form hydrogen cyanide and acetone (Fall, 2003; Gruber et al., 2004). Decarboxylation of acetoacetate is another mechanism that has been hypothesized for acetone production (Macdonald and Fall, 1993).

McKinney et al. (2011) reported several events of negative fluxes of acetone (up to -50 $\mu g\ m^{-2}\ h^{-1}$), especially at night and in the early morning. Several other studies have also reported negative acetone night fluxes at canopy scale (Karl et al., 2004, 2005). Winters et al. (2009) reported a negative correlation between emission rates and ambient concentration that also suggests a bidirectional acetone exchange. Moreover, at leaf level Cojocariu et al. (2004) reported events of acetone uptake during darkness. In principle plant uptake of acetone is a possible explanation. There are very limited literature studies supporting this hypothesis, however. Furthermore, as pointed out by Cojocariu et al., (2004), there is no known mechanism for acetone uptake within plants. In fact, plant fumigation in controlled environments generally does not result in acetone uptake [see e.g. (Omasa et al., 2002)]. Only transient episodes of acetone uptake were reported by Tani and Hewitt (2009) which can be explained by solubility of acetone in the water content



inside the leaf, not followed by metabolization or translocation. Consistent with these results, no episodes of branch-scale acetone deposition were detected during our campaign. Furthermore, in the ancillary experiments, when upper canopy red oak branches were fumigated with acetone, no acetone uptake was detected (Table 3), despite acetone fumigation concentrations as high as ca. 70 ppbv. These results suggest negative canopy-level fluxes of acetone may be due to a mechanism other than stomatal uptake. The vertical profiles (Fig. 2) indicate nighttime acetone concentrations in the range 1-3 ppbv, with a decreasing trend from above to below canopy. As in the case of MEK, this may suggest a ground sink of acetone at night. This hypothesis should be verified in future experiments.

## 3.7 Methanol

Methanol production by vegetation is known to be related to growth and tissue repair processes. It is produced from the demethylation of membrane pectins during formation of cell walls (Fall, 2003; Fall and Benson, 1996). Long term methanol monitoring (Schade and Goldstein, 2006) shows methanol emission peaks during spring in conjunction with the rapid growing of leaves. Hence, growth rate is a key factor controlling plant methanol emission as it strongly influences internal production rate within the plant (Harley et al., 2007). Additional drivers of methanol emissions include temperature and stomatal conductance (Harley et al., 2007). While in the case of acetaldehyde stomata exert long-term control on emission rates (Jardine et al., 2008), stomatal limitations on methanol emissions are short-term. As methanol (as well as other oVOCs) is more soluble in water than non-oxygenated VOCs, it builds up liquid foliar pools that result in a constant flux out of the leaf despite stomatal movements (Niinemets and Reichstein, 2003).





During the campaign, methanol emissions from red oak upper canopy branches were up to 3000 nmol m$^{-2}$ h$^{-1}$ on a leaf area basis (Table 1). Emissions peaked during the day and dropped at night to an average of 300 nmol m$^{-2}$ h$^{-1}$. Scaling up the branch-level data (August 2014) to the ecosystem scale using the same assumptions as for acetone, the resulting daytime (10.00-15.00 h) methanol fluxes range between ca. 200-400 μg m$^{-2}$ h$^{-1}$ on a ground area basis. This is in good agreement with canopy-level flux measurements in the same forest by McKinney et al. (2011), who reported daytime ecosystem-scale fluxes of ca. 100-400 μg m$^{-2}$ h$^{-1}$ on a ground area basis in July 2007 and also observed a similar diel emission pattern. Based on the branch-level experiments, the net 24-hour average methanol exchange was 650 nmol m$^{-2}$ h$^{-1}$ on a leaf area basis or 3400 nmol m$^{-2}$ h$^{-1}$ on a ground area basis. At a Missouri Ozark mixed forest, Seco et al. (2015) measured a net ecosystem flux of ca. 2500 nmol m$^{-2}$ h$^{-1}$ (May through October). Laffineur et al. (2012) studied methanol exchange in the mixed forest of Vielsalm (Belgium) and found a negative average net ecosystem flux (ca. -350 nmol m$^{-2}$ h$^{-1}$; July through October).

No methanol deposition was observed at branch-scale during the campaign. The ancillary experiments (Table 3, Figure 7) show that for the same branches methanol deposition occurs when ambient concentration is rather high. Methanol compensation points in our data were ca. 15-20 ppbv at night and ca. 25-45 ppbv during the day (Figure 7). However, throughout the campaign ambient methanol mixing ratios did not exceed 10 ppbv (Figure 1), which explains why no uptake was observed under ambient conditions. McKinney et al. (2011) reported near zero methanol fluxes at Harvard Forest during night, although methanol uptake events were detected, often in the early morning. Vertical profiles of methanol concentrations (Figure 2) are consistent with a methanol source within the



canopy during daytime while at nighttime the lowest layer displays a significantly lower concentration than the upper layers. This might indicate a lower canopy/ground nighttime methanol sink.

In the fumigation experiments, methanol deposition is not transient and a compensation point is present (Figure 7) suggesting the existence of a methanol degradation mechanism. Likewise, many studies have reported significant canopy-scale methanol deposition to vegetation or soil and evidence of bidirectional exchange (Wohlfahrt et al., 2012), but a holistic mechanistic explanation is lacking. Other studies suggest that methanol is taken

up by stomata. For example, Gout (2000) reported stomatal uptake followed by oxidation of methanol producing formaldehyde. Given the similar uptake rates found during day and night (Figure 7), this would imply that stomata were still (partially) open at night (Niinemets and Reichstein, 2003). As already mentioned, this is corroborated by the fact that the water vapour emissions remain positive at night, although lower than daytime

emissions (data not shown). Another possibility is the consumption of methanol by microorganisms that are commonly found on leaves (Holland and Polacco, 1994) and are able to degrade it via enzymatic reactions, e.g. methylotrophic bacteria (Duine and Frank, 1980). A further hypothesis is chemical transformation of methanol dissolved in water films on leaf surfaces, e.g. via reaction with OH radicals (Elliot and McCracken, 1989).

**3.8 Monoterpenes**

Although isoprene is by far the most abundant isoprenoid emitted by red oaks, monoterpene emission may also be found. In previous branch enclosure studies, Kim et al. (2011) and Helmig et al. (1999) both detected monoterpene emissions from red oaks

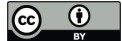



amounting to about 1.1% of total isoprenoids, though with slightly different monoterpene compositions. In our experiment, the daytime monoterpenes/total isoprenoids ratio was in the range of 0.04-0.25% ($\mu$g m$^{-2}$ s$^{-1}$ / $\mu$g m$^{-2}$ s$^{-1}$), thus about an order of magnitude lower than in previous studies.

In the literature, plant species are considered to emit either isoprene or monoterpenes (Harrison et al., 2013), which indicates a possible trade-off of available carbon among isoprenoids (Dani et al., 2015). However, there are important exceptions. First, there are families that actually emit both isoprenoids, although only monoterpenes are stored in permanent pools (Loreto and Fineschi, 2015). Second, emissions of isoprene and monoterpenes in the same plants may occur at different developmental stages, In general, monoterpenes are emitted by young, expanding leaves that upon maturation produce and emit only isoprene. For example, *Picea abies* switches from emitting monoterpenes during May (Schürmann et al., 1993), to predominantly isoprene in June and sesquiterpenes in July (Hakola et al., 2003). Young leaves of several species of poplars (*Populus* sp) emit monoterpenes before complete expansion, when isoprene emission gradually replaces monoterpenes (Brilli et al., 2009; Hakola et al., 1998). Emissions of monoterpenes by red oaks in our branch experiments may therefore indicate the presence of young, growing leaves.

Oaks emitting large amounts of monoterpenes have been described, and are generally evergreen species living in Mediterranean conditions (Loreto et al., 1998, 2009). These oaks do not store monoterpenes in permanent pools and therefore emit monoterpenes in an isoprene-like fashion, with similar light and temperature dependence (Loreto et al., 1996).



We measured daytime emission of monoterpenes with a diurnal cycle very similar to that of isoprene (Figure 1), and a highly significant correlation is present between daytime isoprene and monoterpene emissions (Pearson correlation coefficient 0.88, p<0.05). This suggests *de novo* monoterpene biosynthesis (Loreto et al., 1996), also in a predominantly monoterpene-emitting temperate oak. However, in our experiment monoterpene emissions were also detected during nighttime (Table 1 and Table 2), with average rates of 2-20 nmol $m^{-2}h^{-1}$, suggesting at least some emission from stored pools.

Finally, monoterpene deposition was found on only one out of the three branches sampled (Table 2). By analogy with the much larger emissions of isoprene where deposition was not observed, it may be suggested that monoterpene deposition may be rare or absent.

## 4 Conclusions

The observations presented herein provide new information about the bidirectional exchange of oVOCs between forest plants and the atmosphere. In the predominantly isoprene-emitting red oaks that were the subject of our investigation, we found a link between exchanges of MVK and MEK, which provides a new framework for the understanding of MEK exchange, based on a possible in-plant source from isoprene oxidation products. This link supports the hypothesis of plant stress defense by isoprene via reaction with ROS. Production of MEK in excess of MVK uptake and in correlation with isoprene emission suggests within-plant oxidation of isoprene to MVK followed by detoxification and the eventual release of volatile products such as MEK, 3-buten-2-ol and 2-butanol. Further studies are needed to confirm this link. We also found small emissions of monoterpenes, which might be a marker of juvenility in the canopy. We report





bidirectional exchange of benzaldehyde between biosphere and the atmosphere, describing a so far unknown (to our knowledge) foliar deposition of benzenoid compounds. More investigations on benzenoid bidirectional exchange are needed to improve global budgets of the biogenic and anthropogenic sources of these volatiles. Emissions and fluxes of other

important oVOCs such as acetaldehyde, methanol and acetone have been confirmed.

**Data availability**

The full dataset is available from the authors upon request (email). Data will also be available on the Harvard Forest archive.

*Author contributions*. L. Cappellin and K. McKinney designed this research; field work was carried out by L. Cappellin, A. Algarra Alarcon, I. Herdlinger-Blatt and K. McKinney. Laboratory analyses were performed by A. Algarra Alarcon and L. Cappellin. L. Cappellin, F. Biasioli, S. Martin, F. Loreto and K. McKinney wrote the paper.

*Competing interests*. The authors declare that they have no conflict of interest.

*Acknowledgements.* Luca Cappellin acknowledges funding from H2020-EU.1.3.2 (grant agreement n. 659315). The authors thank Evan Goldman for providing LAI data and Mark Vanscoy for providing meteorological data and for managing the measurement site.





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



# Tables

**Table 1.** Average measured emission and deposition rates at branch-scale in the upper canopy. August 14 - 25. Data are reported as mean ± standard deviation. Maximum
5 emission and deposition rates are also reported as well as an indication of their statistical significance (*: p<0.05; n.s.: not significant).

|  | Average daytime emission | Average nighttime emission | Average 24-hour emission | Maximum emission rate (1 h integration) | Maximum deposition rate (1 h integration) |
|---|---|---|---|---|---|
|  | [nmol/m$^2$/h] | [nmol/m$^2$/h] | [nmol/m$^2$/h] | [nmol/m$^2$/h] | [nmol/m$^2$/h] |
| Benzaldehyde | 0.6 ± 0.3 | 0.2 ± 0.1 | 0.4 ± 0.2 | 2.2 * | -1.9 * |
| Acetaldehyde | 50 ± 30 | 11 ± 6 | 20 ± 10 | 170 * | -20 * |
| MEK | 30 ± 10 | 4 ± 3 | 15 ± 7 | 140 * | 0 |
| Acetone | 90 ± 50 | 20 ± 30 | 50 ± 30 | 330 * | 0 |
| MeOH | 1000 ± 800 | 300 ± 100 | 600 ± 400 | 3000 * | 0 |
| ISOP | $0.2 ± 0.1 \times 10^5$ | $0.0 ± 0.0 \times 10^5$ | $0.10 ± 0.07 \times 10^5$ | $1.0 \times 10^5$ * | $0.0 \times 10^5$ |
| Monoterpenes | 20 ± 10 | 2 ± 2 | 10 ± 7 | 183 * | 0 |
| MACR+MVK+ISOPOOH | - | 0 ± 2 | - | 7 n.s. | -4 * |





**Table 2.** Average measured emission and deposition rates at branch-scale in the upper canopy. August 25 - September 1. Data are reported as mean ± standard deviation. Maximum emission and deposition rates are also reported as well as an indication of their statistical significance (*: $p<0.05$; n.s.: not significant).

|  | Average daytime emission | Average nighttime emission | Average 24-hour emission | Maximum emission rate (1 h integration) | Maximum deposition rate (1 h integration) |
|---|---|---|---|---|---|
|  | [nmol/m²/h] | [nmol/m²/h] | [nmol/m²/h] | [nmol/m²/h] | [nmol/m²/h] |
| Benzaldehyde | -1 ± 2 | 0 ± 1 | -1 ± 2 | 0.5 * | -7 * |
| Acetaldehyde | 10 ± 40 | -1 ± 10 | 3 ± 18 | 100 * | -40 * |
| MEK | 60 ± 20 | 7 ± 7 | 30 ± 20 | 460 * | 0 |
| Acetone | 50 ± 20 | 10 ± 10 | 30 ± 20 | 340 * | 0 |
| MeOH | 1000 ± 1000 | 300 ± 200 | 700 ± 600 | 3000 * | 0 |
| ISOP | 0.4 ± 0.4 x10⁵ | 0.0 ± 0.0 x10⁵ | 0.2 ± 0.2 x10⁵ | 3.2 x10⁵ * | 0.0 x10⁵ |
| Monoterpenes | 50 ± 50 | 20 ± 10 | 40 ± 40 | 1370 * | -50 n.s. |
| MACR+MVK+ISOPOOH | - | -6 ± 8 | - | 0 | -23 * |

**Table 3.** Summary of the ancillary experiments: Fumigation of forest red oak upper canopy branches with VOCs.

|  | Henry Law Constant (water/air) $H^{cp}$ at $T^{\Theta}$ [mol / (m³·Pa)] | VOC fumigation concentration range [ppbv] | Daytime deposition | Nighttime deposition |
|---|---|---|---|---|
| MVK | 2.6·10⁻¹ | 0 - 70 | YES | YES |
| MACR | 4.8·10⁻² | 0 - 50 | YES | YES |
| ISOP | 1.3·10⁻⁴ | 0 - 700 | - | - |
| ACETONE | 2.7·10⁻¹ | 0 - 70 | - | - |
| MeOH | 2.0 | 0 - 70 | YES | YES |



## Figure captions

**Figure 1.** Diel average measured mixing ratios and fluxes in enclosures containing upper canopy red oak branches at Harvard Forest. Mixing ratio of the inflow ambient air and of the outflow air are shown in red and black, respectively. Diel average temperature and PAR during the measurement period are also shown. Error bars represent standard errors of the data points in each time interval.

**Figure 2.** Average daytime and nighttime vertical concentration profiles. Measured VOC concentrations at various heights within and above canopy are reported. Horizontal bars represent standard deviations.

**Figure 3.** Example of branch-level VOC flux measured as a function of MVK fumigation mixing ratio during daytime. The concomitant MVK uptake and emission of MVK reduction products (namely MEK, 3-buten-2-ol and 2-butanol) implies plant detoxification of MVK.

**Figure 4.** Scatter plot of branch-level MEK fluxes versus MACR+MVK+ISOPOOH fluxes during darkness in the period 21-25 August.

**Figure 5.** Example of benzaldehyde daily pattern for a day when benzaldehyde emission was detected. Measurements correspond to three different red oak upper canopy branches.

**Figure 6.** Flux measured as a function of mixing ratio for benzaldehyde during daytimes of the whole branch-level measurement campaign. Black triangles represent the period 25 August - 1 September, when deposition dominated, and grey dots correspond to the period 14-25 August, where mostly emission was measured.

**Figure 7.** Example of enclosure flux measured as a function of fumigation mixing ratio for methanol in the field.



**Fig01**





**Fig02**



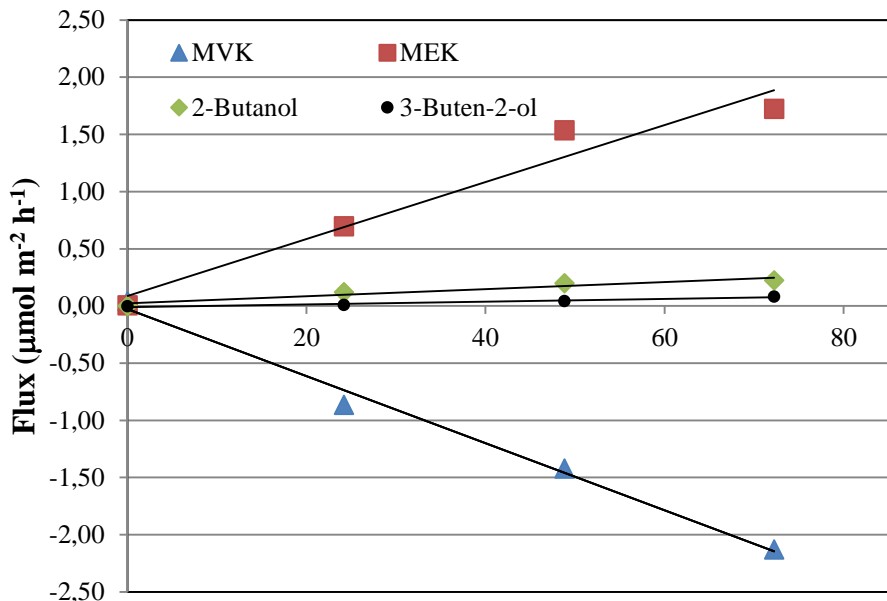

**MVK Fumigation Mixing ratio (ppbv)**

**Fig03**



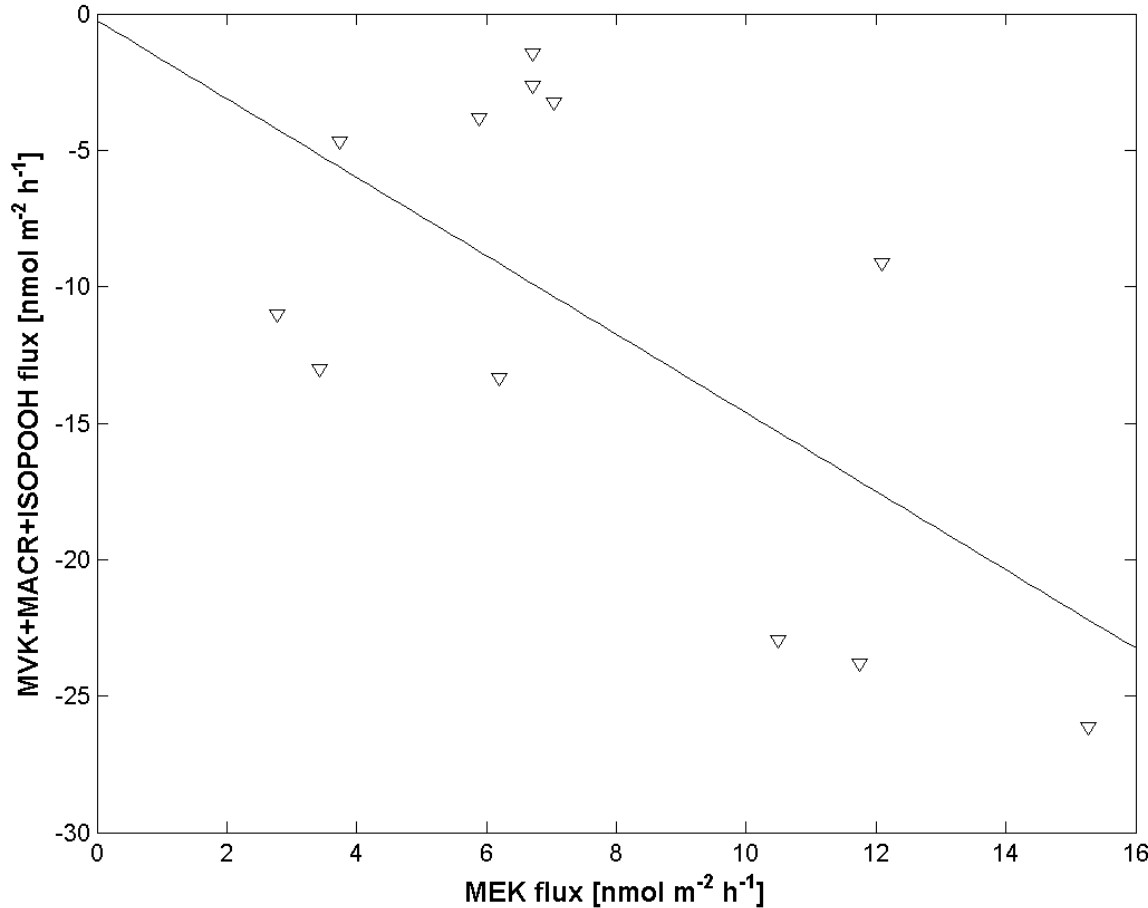

**Fig04**





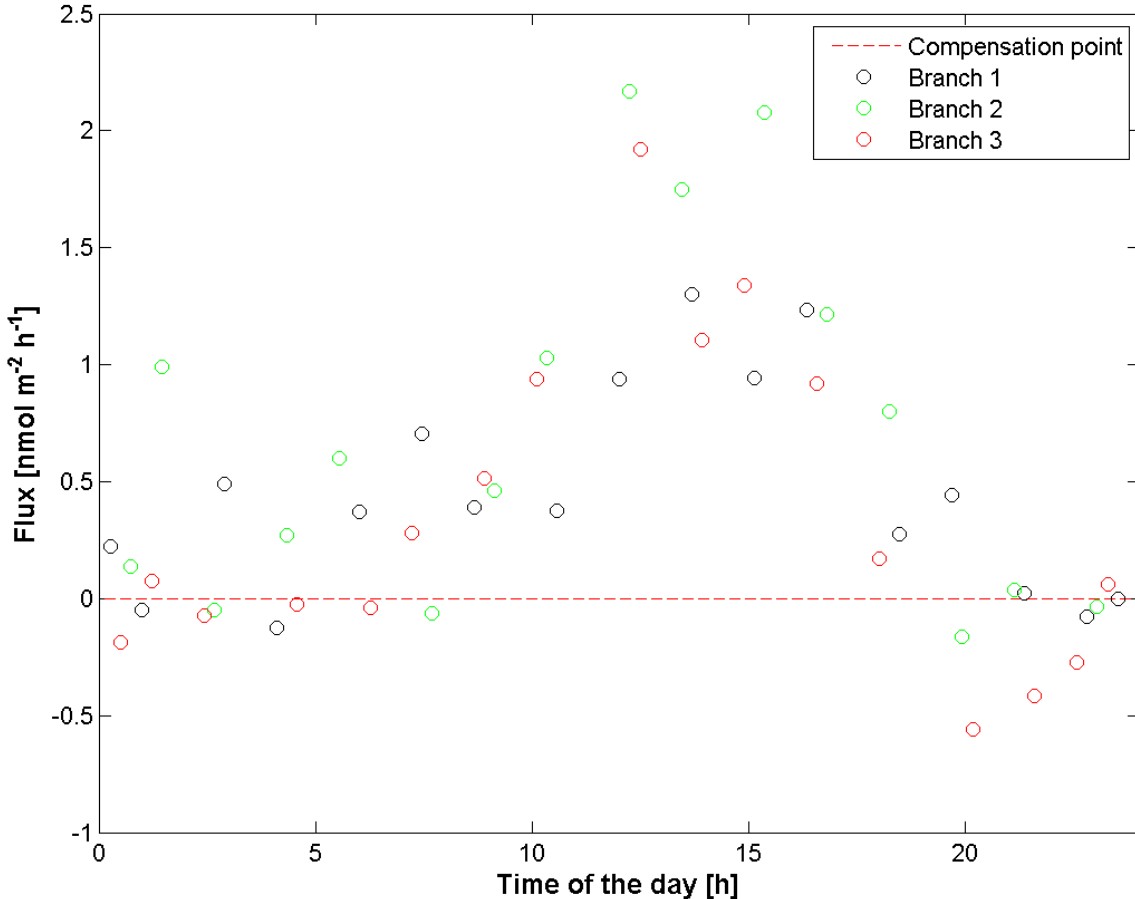

**Fig05**




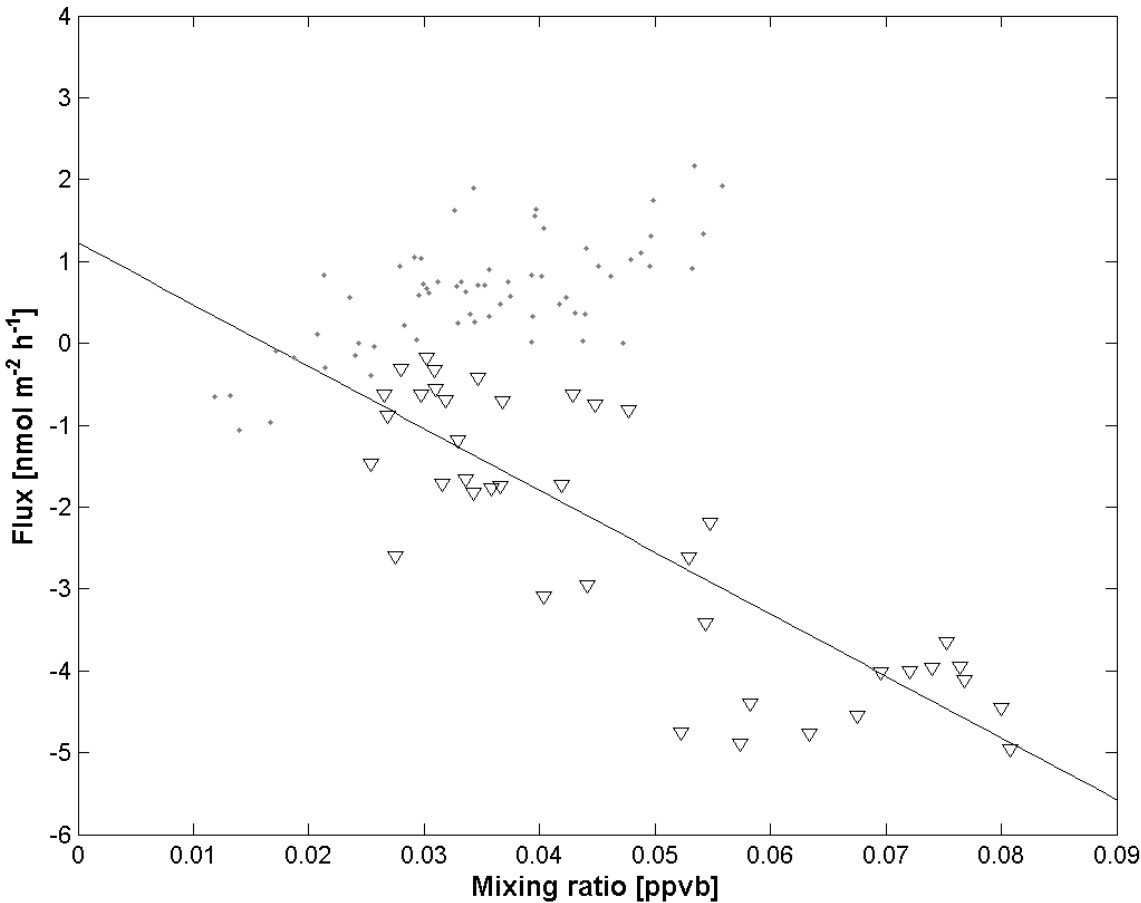

**Fig06**




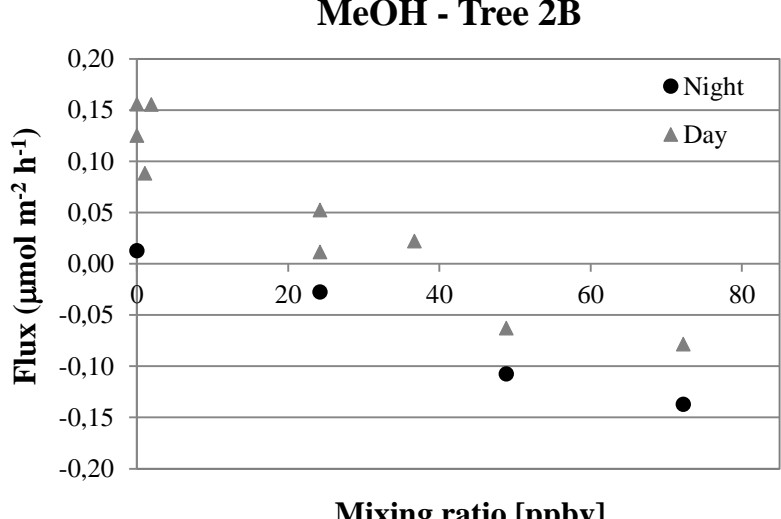

**Fig07**