# Peer review of "Field observations of Volatile Organic Compound (VOC) exchange in red oaks"

_Atmospheric Chemistry and Physics, 2016_

## Referee Comment (RC1) · Anonymous Referee #1 · 21 Nov 2016

**Field observations of Volatile Organic Compound (VOC) exchange in red oaks**

General comments:
Cappellin and coauthors present 42 days of VOC measurements above a mixed red oak forest in central Massachusetts. The measurements consisted of 22 days of concentration profile, followed by 20 days of branch chamber measurements, where the last day was used for fumigation. In the results 6 different VOCs and 2 VOC groups are presented. The manuscript meets the scope of the journal and can be accepted for publication after addressing following comments:

Specific comments:

Did the authors present all VOCs that showed exchange? If not, how many were measured and what was the selection criteria?

P6 L14: as it is a mixed forest: what are the major tree species? This information can be helpful for the interpretation of the concentration profiles and the flux upscaling.

P7 L21 & P9 L9: randomly cycle between the inlets -> what are the benefits of randomly cycling? Isn't it disadvantageous, that e.g. the highest inlet could be measured for 1 h in a row, while a lower inlet would be measured with a 1:40 h gap? Please rephrase so it is more clear that random cycling means that all four heights were sampled in a random order during 1 h of measurements (and not, that random means that e.g. one height could be measured multiple times during 1 h).

P9 L24 – P10 L5: The fumigation experiments are not well described here, please specify:
- in which mode was the instrument used (according to P15 L20 it was $NO^+$, but it is not clear if also $H_3O^+$ was used),
- the used concentrations (in the gas cylinders)
- the target VOCs
- what was the final concentration or dilution which was delivered to the branch enclosure?
- how long did it take (in average) until the signal was stable?

This information is important for the reader.

P10 L7: 2.4 PTR/SRI-ToF-MS operation and data analysis: Please state typical primary ion counts for $H_3O^+$ and $NO^+$ mode and add the percentage of the first water cluster (in the $H_3O^+$ mode).

P13 L9-13: What were the uncertainties of the calibration gas? What were typical concentrations/dilution used for calibrations and could the authors state the used sensitivities (measured and calculated)?

P13 L18: Please reformate this sentence, as the reader could get the impression that the uncertainty of the calculated sensitivity is 10%. While I guess the author means that the calculated sensitivity is within 10% of the experimental estimated sensitivity (which has a 15% uncertainty). Otherwise the reader could believe that calculating a sensitivity is actually better (less uncertainty) then getting it via calibration. Please also state that the cited study was measured in a laboratory with known compounds and known fragmentation patterns. Whereas here the chemical structure is not known (e.g. monoterpenes) and fragmentation patterns vary. Additionally fragments cannot be assigned to one compound, as many compounds fragment to the same masses (e.g. $C_3H_5^+$; $C_2H_3O^+$; $C_6H_9^+$).
Please remove the "Mueller et al., 2014" reference, as it just very briefly names the theoretical approach, but doesn't compare calculated with theoretical sensitivities.

P15 L19: It is true that 2-butanol is measured in the $H_3O^+$ mode at M 57.070 ($C_4H_9^+$). But there could be a potential influence from protonated butene, which is known to be emitted from trees (e.g. Hakola et al., 1998). Was this considered (and dismissed by comparison with the $NO^+$ measurements)? Were there any interferences from the water cluster ($H_7O_3^+$) when measuring 3-buten-2-ol in the $H_3O^+$ mode?

P17 L13: Have there been more studies which showed ecosystem MVK + MACR emissions from oak forests since 2005?

Technical corrections:

P3 L3 & following: order of references: chronological; here: (Guenther et al., 1995; Benkovitz et al., 2004). (The references at the end of the manuscript are sorted alphabetically).

P3 L8: forming -> to the formation of

P4 L21: quite toxic -> toxic

P7 L3: NOy -> $NO_y$ (y subscript)

P7 L14 & following: inch –> metric units (SI units), please change all imperial units to SI units.
  Please have a look at: http://www.atmospheric-chemistry-and-physics.net/for_authors/manuscript_preparation.html
  at the "Mathematical notation and terminology"

P7 L18 & following: slm -> L min$^{-1}$

P8 L3-4: remove bold format

P8 L8: (See Section 2.4 for details) -> (see Sect. 2.4)

P8 L9: Palladium –> palladium

P8 L18: [version 1.47, (Schneider et al., 2012)] -> (version 1.47; Schneider et al., 2012)

P8 L18: A fourth bag not containing a branch was -> A fourth, empty enclosure was

P8 L24: Bios DryCal Flow Calibrators -> please specify the used type

P9 L4: to achieve residence times of 38s -> please specify that the residence time belongs to the enclosure

P9 L17: in the empty bag -> empty enclosure

P9 L21: (See Section 2.4 for details) -> (see Sect. 2.4)

P10 L1: flow controllers (MKS instruments) -> please do not state just a company, always add the used type of instrument

P10 L20-21: …per channel …..350,000 channels -> "bin" is a more common used word (e.g. Graus et al., 2010) as channels could be confused with the channels in the MCP; -> 350 000 in the remaining manuscript not thousand-separator was used (see Table 1 & 2; Fig 01)

P11 L5 & following: (5a), (5b),… please start with 1 (not 5) and proceed from there. For chemical reactions (R1), (R2),.. is used.

P11 L24 & following: (channel 5b) -> (Eq. R2)

P16 L4 & following: Figure 1 -> Fig. 1 (Figure X is just used when starting a sentence, otherwise Fig. X is used)

P28 L23-24: [see e.g. (Omasa et al., 2002)] -> (e.g. Omasa et al.,2002)

P32 L11: stages, In -> stages. In

Table 1 &2: [nmol/m$^2$/s$^1$] -> [nmol m$^{-2}$ s$^{-1}$]
  please change the average daytime emission of ISOP to $2\pm1$ x10$^4$ (same for average 24h emission)
  if there was no isoprene flux during night, please change $0.0 \pm 0.0$ x10$^5$ to 0; if there was a flux then please give proper values (same goes for maximum deposition rate)

Table 3: [mol / (m$^3$·Pa)] -> [mol m$^{-3}$ Pa$^{-1}$]

Fig03 & Fig07: please use same notation for axis label units []

Fig 03,04,06: state the fit parameters in the figure (legend)

References:

Graus, M., Müller, M., and Hansel, A.: High resolution PTR-TOF: quantification and formula confirmation of VOC in real time, J. Am. Soc. Mass Spectr., 21, 1037–1044, 2010.

Hakola, H., Rinne, J. and Laurila, T.: The hydrocarbon emission rates of tea-leafed willow (Salix phylicifolia), silver birch (Betula pendula) and European aspen (Populus tremula), Atmos. Environ., 32(10), 1825–1833, doi:10.1016/S1352-2310(97)00482-2, 1998.

Müller, M., Mikoviny, T. and Wisthaler, A.: Detector aging induced mass discrimination and non-linearity effects in PTRToF-MS, Int. J. Mass Spectrom., 365–366, 93–97, doi:10.1016/j.ijms.2013.12.008, 2014.

---

## Referee Comment (RC2) · Anonymous Referee #2 · 6 Dec 2016

The manuscript of Cappellin et al. reports fluxes of volatile organic compounds (VOC) from branch enclosure and canopy profile measurements in a mixed forest. Moreover, the author performed branch-level fumigation experiments by employing some selected Oxygenated-VOCs to get new insights into the sources and sinks of those VOCs. The manuscript is generally well written and fits the scope of the journal. However, I suggest the author to address some major and minor issues (listed below) to get the manuscript ready for publication.

MAJOR ISSUES: - The author collected data over a period spanning 2 weeks (from 14th of August to 1st September) which can be defined as a 'short-term' period of time. Therefore, the author should take into account that emission/deposition rates of (O)VOC and their dynamics presented in this manuscript can change over a longer period of time (i.e. months). - Despite the concentration of several VOCs have been

measured simultaneously by PTR-TOF-MS, the author reported in the manuscript only "the most important OVOC and volatile isorpenoids" (as written in line 1, page 16). The author should clearly define 'how' those oVOC and the isoprenoids have been selected/filtered form the multitude of protonated ions related to VOC (and/or fragment of VOC.) recorded by PTR-TOF-MS. - I remind the author that 'concentrations' of VOC shown in Figure 1, may represent a larger area possibly including other sources/sinks than the mixed forest investigated by the author. Indeed only through a 'footprint analysis', a model that considers the concentration distribution of VOC with respect to wind speed and the turbulent air conditions occurring at the site, it can be sorted out the representativeness of the measured concentration on the surface area (Horst and Weil 1992, 1994; see also Schmid 1994 for details). - The author discussed the results, i.e. of methanol fluxes (lines 9-24, page 30) by comparing VOC fluxes calculated in this manuscript from measurements collected at branch-level (and then scaled-up at canopy-level after considering the average LAI and inhomogeneity of the canopy; as written in lines 20-24, page 16 and lines 1-4, page 17), with results found in the literature where VOC fluxes have been measured at canopy level by eddy covariance technique. The author should discuss and make the point on how the different approach in calculating the VOC fluxes may have produced different results (i.e. eddy covariance flux measurements provide emission/deposition fluxes from complex sources/sinks also including the soil, while branch measurements provide indications of VOC emissions mainly form leaves as the only source/sink). - I am wondering whether the author can provide additional data on both the $CO_2$ and water vapor exchange from branch enclosure measurements. Indeed, the measurements of $CO_2$ assimilation rates would give information on the physiological performances of the leaves enclosed in the branch chamber; this would be particularly important to evaluate any damage eventually occurring to primary leaves metabolism following MVK absorption (in correlation with detoxification and thus MEK emission). In addition, measurements of water vapor exchange (even though at branch level) would provide basic information useful to both estimate the extent of stomata conductance and to partition the stomata and

non-stomatal contribution to the uptake of (O)VOC (Fares et al. 2012) that has been mentioned several times throughout the manuscript. - Since the 'compensation point' may vary on the basis of both physiological and physicochemical factors (as reviewed by Niinemets et al. 2014), when discussing this issue it would be better to indicate, at least, the environmental conditions under which this point has been determined. - Material and Methods section needs to be re-formatted and dramatically shorten it (now it is almost 10 pages long!). I suggest the author to either delete or move to 'Supplementary Information' most of the description of PTR-TOF-MS technical details (which is now 6 pages long) and simply mention citations of the many previous scientific works where PTR-TOF-MS has been already decribed. In fact, subsections 2.4.1 and 2.4.2 belong more to a textbook than to a scientific paper, while much of the 'Spectral analysis' subsection could be summarized in a table (i.e. by comparing VOC analysys by PTR-TOF-MS set in H3O+ and NO+ mode). Moreover, if not commercial, a schematic and/or a picture of the canopy-top branch enclosure employed for the measurements would be very much informative.

MINOR ISSUES: - Consistency I required throughout the manuscript! Please make use of either the term BVOC or VOC. - Is there any particular reason why fluxes are always expressed in "nmol m-2 hour-1", instead of "nmol m-2 second-1"? (in which all the measurements unit are indicated according to the International System of Units) - Lines 12-13, page 1: I suggest to replace 'atmospheric reactor' with 'atmosphere'. - Line 14, page 6: the author should describe the plant species composition of the mixed forest that has been investigated; this can offer information to improve the discussion of sources/sinks of (O)VOC emission/deposition. - Line 21, page 6: it is written 'Influence', without specifying 'of what'? - Lines 3-4, page 8: I wonder why the fonts of this sentence are in 'bold'... - Lines 14-15, page 16: This is not clear, and I suggest the author to rephrase it. - Lines 13-14, page 17: I remind the author that also Brilli et al. (2016) showed MVK+MAC emission fluxes at canopy-level from a poplar plantation. - Line 4, page 18: the author should quickly explain 'why' 'the actual presence of emissions could not be proved. - Lines 4-5, page 20: I suggest to remove both the

words 'first' and to replace the last word with 'before'. - Lines 14-17, page 20: I remind the author that also Brilli et al. (2016) found emission and deposition fluxes of MEK through eddy covariance measurements at canopy level. - Lines 19-20, page 22: the author should specify which kind of 'correlation' (i.e. linear?) - Lines 1-3, page 23: I remind the author that emission of alcohols (i.e. belonging to the mixture of Green Leaf Volatiles) does not depend to the plant species, but to the occurrence of specific situation (i.e. the presence of herbivores inducing mechanical damage to leaves). - Line 16-17, page 24: the author meant 'benzaldehyde concentration', in air ? - Lines 5-6, page 27: These citations are missing in the reference list. - Lines 7-8, page 28: I am wondering if the Model of Emissions of Gases and Aerosols from Nature has been never applied to simulate acetone emissions. - Line 18, page 29: I remind the author that also Hüve et al. 2007 investigated the control of methanol release by leaf expansion and stomata.

---

## Author Response (AR1)

**Dear Editor,**

We thank the two reviewers for their thoughtful and detailed comments on our manuscript. Please find below our response to the comments raised by the reviewers. We hope to have satisfactorily responded to all reviewer suggestions and comments, which have substantially improved the manuscript. Changes in the revised manuscript were made using the track-mode tool and are visible. Author comments below are in bold.

**REVIEWER #1:**

General comments:

Cappellin and coauthors present 42 days of VOC measurements above a mixed red oak forest in central Massachusetts. The measurements consisted of 22 days of concentration profile, followed by 20 days of branch chamber measurements, where the last day was used for fumigation. In the results 6 different VOCs and 2 VOC groups are presented. The manuscript meets the scope of the journal and can be accepted for publication after addressing following comments: **We thank the reviewer for these comments**.

Specific comments:

Did the authors present all VOCs that showed exchange? If not, how many were measured and what was the selection criteria?

Although VOCs are detected at many m/z ratios, we focused on VOCs that showed significant ecosystem-level fluxes in previous measurements at Harvard Forest (McKinney et al., 2011). In addition, we report benzaldehyde since, to the best of our knowledge, deposition of this compound has never been reported in forest so far.

At page 11, lines 5-7, the following sentence has been added:

"The reported compounds were selected based on canopy-scale flux measurements previously performed at a nearby site (McKinney et al., 2011) with the addition of benzaldehyde."

P6 L14: as it is a mixed forest: what are the major tree species? This information can be helpful for the interpretation of the concentration profiles and the flux upscaling.

The major tree species present in the mixed forest can be found in Urbanski et al. (2007). At page 6, lines 19-23, the following sentence has been added:

"A survey of the tree species present at EMS is reported in Urbanski et al. (2007): The mixed forest stand encompasses red oak (36 % of biomass), red maple (22 %), hemlock (13 %), red pine (8 %), birch (5 %), white pine (6 %), cherry (3 %), spruce (2 %) and beech (0.8 %). The species distribution at the walk-up tower site is similar but not identical."

P7 L21 & P9 L9: randomly cycle between the inlets -> what are the benefits of randomly cycling? Isn't it disadvantageous, that e.g. the highest inlet could be measured for 1 h in a row, while a lower inlet would be measured with a 1:40 h gap? Please rephrase so it is more clear that random cycling means that all four heights were sampled in a random order during 1 h of measurements (and not, that random means that e.g. one height could be measured multiple times during 1 h).

We thank the reviewer for noting this point. We rephrased the two sentences in order to clarify that random cycling means that all four heights were sampled in a random order during 1 h of measurements.

At page 8, lines 1-3, the sentence now reads:

"The valve system was set to cycle between the inlets every 10 min (30 min in the case of tower top inlet) so that all inlets were measured in random order every 60 min."

At page 9, lines 12-15, the sentence now reads:

**"Each branch enclosure was sampled for 10 min, followed by 10 minutes from the empty enclosure. The valve system was set to cycle among the branch enclosures so that all of them were measured in random order"**

P9 L24 – P10 L5: The fumigation experiments are not well described here, please specify: - in which mode was the instrument used (according to P15 L20 it was NO+, but it is not clear if also H3O+ was used),

- the used concentrations (in the gas cylinders)

- the target VOCs

- what was the final concentration or dilution which was delivered to the branch enclosure?

- how long did it take (in average) until the signal was stable?

This information is important for the reader.

We added the suggested information to the experimental description (page 10, lines 6-19):

"On September 2 ancillary fumigation experiments were carried out. Gas cylinders containing known amounts of the target VOCs (Scott Specialty Gases, Inc.) were diluted with zero air using mass flow controllers (MKS Instruments) and delivered to the branch enclosures. Tested VOCs included isoprene (80 ppm  $\pm$  5% in the gas cylinder), MVK (1 ppm  $\pm$  5%), MACR (1 ppm  $\pm$  5%), acetone (1 ppm  $\pm$  5%), and methanol (1 ppm  $\pm$  5%). For each VOC different concentrations in the range 0-70 ppbv (0-700 for isoprene) were delivered to the branch enclosures. Flow rates were set as previously described. For each concentration step, the enclosures, including the empty one, were measured in random order and the signal was allowed to stabilize (typically for tens of minutes) before a measurement was made. The PTR/SRI-ToF-MS was operated in NO+ mode. VOC exchange rates were computed as described above."

P10 L7: 2.4 PTR/SRI-ToF-MS operation and data analysis: Please state typical primary ion counts for H3O+ and NO+ mode and add the percentage of the first water cluster (in the H3O+ mode). Typical primary ion counts were  $1.1 \cdot 10^6$  cps for H3O+ mode and  $1.2 \cdot 10^6$  cps for NO+ mode. The percentage of first water cluster in the H3O+ mode was ca. 19%.

In the supplement, at page 1, lines 10-11, the following sentence was added: "The primary ion signal was  $1.1 \cdot 10^6$  cps and the percentage of first water clusters was about 19%."

In the supplement, at page 1, lines 13-14, the following sentence was added: "The primary ion signal was 1.2·106 cps".

P13 L9-13: What were the uncertainties of the calibration gas? What were typical concentrations/dilution used for calibrations and could the authors state the used sensitivities (measured and calculated)?

We added the required specifications.

In the supplement, at page 3, lines 23-25, the following sentences were added: "The uncertainties of the calibration gases were ± 5%. Typical concentrations used for instrumental calibration were in the range 1-70 ppbv. For isoprene concentrations used for instrumental calibration were in the range 1 ppbv - 1 ppmv."

In the supplement, at page 4, lines 7-15, the following sentences were added:

"In H3O+ mode the instrumental sensitivities were the following: 16.4 ncps/ppbv for isoprene, 27.1 ncps/ppbv for methacrolein, 28.7 ncps/ppbv for MVK, 28.0 ncps/ppbv for MEK, 40.0 ncps/ppbv for benzaldehyde, 21.4 ncps/ppbv for acetaldehyde, 25.8 ncps/ppbv for acetone, 13.2 ncps/ppbv for methanol, 28.5 ncps/ppbv for monoterpenes; where ncps are counts per second normalized to a primary ion signal of 106 cps. In NO+ mode the instrumental

**sensitivities were the following: 17.6 ncps/ppbv for isoprene, 16.4 ncps/ppbv for methacrolein, 21.0 ncps/ppbv for MVK, 26.5 ncps/ppbv for MEK, 43.7 ncps/ppbv for benzaldehyde, 13.3 ncps/ppbv for acetone, 29.9 ncps/ppbv for monoterpenes."**

P13 L18: Please reformate this sentence, as the reader could get the impression that the uncertainty of the calculated sensitivity is 10%. While I guess the author means that the calculated sensitivity is within 10% of the experimental estimated sensitivity (which has a 15% uncertainty). Otherwise the reader could believe that calculating a sensitivity is actually better (less uncertainty) then getting it via calibration. Please also state that the cited study was measured in a laboratory with known compounds and known fragmentation patterns. Whereas here the chemical structure is not known (e.g. monoterpenes) and fragmentation patterns vary. Additionally fragments cannot be assigned to one compound, as many compounds fragment to the same masses (e.g. C3H5+; C2H3O+; C6H9+). Please remove the "Mueller et al., 2014" reference, as it just very briefly names the theoretical approach, but doesn't compare calculated with theoretical sensitivities.

We thank the reviewer for this suggestion. We changed the text accordingly. In the supplement, at page 4, lines 1-7, we replaced this sentence

"This method was shown to estimate concentrations within a 10% uncertainty, under certain instrumental conditions (Cappellin et al., 2012)", with the following sentences: "Instrument sensitivities calculated with such method have been shown to agree with sensitivities determined experimentally within 10%, under certain instrumental conditions (Cappellin et al., 2012). In particular this is true when the identity of the compound and the fragmentation pattern are known. In some cases the chemical structure or fragmentation pattern is not known (e.g. monoterpenes) thus the expected uncertainty increases. Moreover, some fragments can arise from multiple compounds (e.g. C3H5+)."

P15 L19: It is true that 2-butanol is measured in the H3O+ mode at M 57.070 (C4H9+). But there could be a potential influence from protonated butene, which is known to be emitted from trees (e.g. Hakola et al., 1998). Was this considered (and dismissed by comparison with the NO+ measurements)? Were there any interferences from the water cluster (H7O3+) when measuring 3-buten-2-ol in the H3O+ mode?

Previous studies on red oak emissions (Helmig et al., 1999) have not reported butene (to the best of our knowledge), therefore we ruled out this potential interference. The paper mentioned by the reviewer did report butene emission but from tree species other than red oaks, namely from tea-leafed willow, silver birch and European aspen. The spectral peak corresponding to the water cluster and the one corresponding to the signal of buten-2-ol partially overlap. The peak deconvolution algorithm was able to separate them although some influence cannot be ruled out completely. There is no significant correlation between the two signals (p>0.05), thus suggesting that the influence is not strong.

P17 L13: Have there been more studies which showed ecosystem MVK + MACR emissions from oak forests since 2005?

We thank the reviewer for noting it. There are actually more recent studies about positive ecosystem-scale fluxes of MVK and MACR in oak forests (Kalogridis et al., 2014; Schallhart et al., 2016). We added such references at page 12, line 19.

**Technical corrections:**

P3 L3 & following: order of references: chronological; here: (Guenther et al., 1995; Benkovitz et al., 2004). (The references at the end of the manuscript are sorted alphabetically).

We chose to order the in-text citations alphabetically and we propose to keep this choice. Citing from the ACP guidelines: "In terms of in-text citations, the order can be based on relevance, as well as chronological or alphabetical listing, depending on the author's preference."

P3 L8: forming -> to the formation of

This point has been corrected.

P4 L21: quite toxic -> toxic

This point has been corrected.

P7 L3: NOy -> NOy (y subscript)

**This point has been corrected.**

P7 L14 & following: inch -> metric units (SI units), please change all imperial units to SI units. Please have a look at: http://www.atmospheric-chemistry-andphysics.net/for\_authors/manuscript\_preparation.html at the "Mathematical notation and terminology"

**This point has been corrected.**

P7 L18 & following: slm -> L min-1

**This point has been corrected.**

P8 L3-4: remove bold format

This point has been corrected.

P8 L8: (See Section 2.4 for details) -> (see Sect. 2.4)

**This point has been corrected.**

P8 L9: Palladium -> palladium

**This point has been corrected.**

P8 L18: [version 1.47, (Schneider et al., 2012)] -> (version 1.47; Schneider et al., 2012)

**This point has been corrected.**

P8 L18: A fourth bag not containing a branch was -> A fourth, empty enclosure was

**This point has been corrected.**

P8 L24: Bios DryCal Flow Calibrators -> please specify the used type

This point has been corrected.

**At page 9, line 5, we specified that we used "Bios Defender 520-H and 520-L".**

P9 L4: to achieve residence times of  $38s \rightarrow$  please specify that the residence time belongs to the enclosure

**This point has been corrected.**

P9 L17: in the empty bag -> empty enclosure

**This point has been corrected.**

P9 L21: (See Section 2.4 for details) -> (see Sect. 2.4)

**This point has been corrected.**

P10 L1: flow controllers (MKS instruments) -> please do not state just a company, always add the used type of instrument

**This point has been corrected.**

**At page 10, line 9, we specified that we used "Mass-Flo Controller, MKS instruments".**

P10 L20-21: ...per channel .....350,000 channels -> "bin" is a more common used word (e.g. Graus et al., 2010) as channels could be confused with the channels in the MCP; -> 350 000 in the remaining manuscript not thousand-separator was used (see Table 1 & 2; Fig 01)

**This point has been corrected.**

P11 L5 & following: (5a), (5b),... please start with 1 (not 5) and proceed from there. For chemical reactions (R1), (R2),.. is used.

**This point has been corrected.**

P11 L24 & following: (channel 5b) -> (Eq. R2)

**This point has been corrected.**

P16 L4 & following: Figure 1 -> Fig. 1 (Figure X is just used when starting a sentence, otherwise Fig. X is used)

**This point has been corrected.**

P28 L23-24: [see e.g. (Omasa et al., 2002)] -> (e.g. Omasa et al., 2002)

This point has been corrected.

P32 L11: stages, In -> stages. In

**This point has been corrected.**

Table 1 &2: [nmol/m2/s1] -> [nmol m-2 s-1]

**This point has been corrected.**

please change the average daytime emission of ISOP to  $2\pm 1 \times 104$  (same for average 24h emission) if there was no isoprene flux during night, please change  $0.0 \pm 0.0 \times 105$  to 0; if there was a flux then please give proper values (same goes for maximum deposition rate)

**This point has been corrected.**

Table 3:  $[mol / (m3 \cdot Pa)] \rightarrow [mol m-3 Pa-1]$

This point has been corrected.

Fig03 & Fig07: please use same notation for axis label units []

The units are different because in one case flux and the other case concentrations are represented.

Fig 03,04,06: state the fit parameters in the figure (legend)

**The fit parameters have been added in the legends of Fig. 03, 04 and 06.**

**References:**

Graus, M., Müller, M., and Hansel, A.: High resolution PTR-TOF: quantification and formula confirmation of VOC in real time, J. Am. Soc. Mass Spectr., 21, 1037–1044, 2010. Hakola, H., Rinne, J. and Laurila, T.: The hydrocarbon emission rates of tea-leafed willow (Salix phylicifolia), silver birch (Betula pendula) and European aspen (Populus tremula), Atmos. Environ., 32(10), 1825–1833, doi:10.1016/S1352-2310(97)00482-2, 1998.

Müller, M., Mikoviny, T. and Wisthaler, A.: Detector aging induced mass discrimination and nonlinearity effects in PTRToF-MS, Int. J. Mass Spectrom., 365–366, 93–97, doi:10.1016/j.ijms.2013.12.008, 2014.

**REVIEWER #2:**

The manuscript of Cappellin et al. reports fluxes of volatile organic compounds (VOC) from branch enclosure and canopy profile measurements in a mixed forest. Moreover, the author performed branch-level fumigation experiments by employing some selected Oxygenated-VOCs to get new insights into the sources and sinks of those VOCs. The manuscript is generally well written and fits the scope of the journal. However, I suggest the author to address some major and minor issues (listed below) to get the manuscript ready for publication.

**We thank the reviewer for these comments.**

**MAJOR ISSUES:**

- The author collected data over a period spanning 2 weeks (from

14th of August to 1st September) which can be defined as a 'short-term' period of time. Therefore, the author should take into account that emission/deposition rates of (O)VOC and their dynamics presented in this manuscript can change over a longer period of time (i.e. months).

The reviewer is right in pointing out that fluxes can change over time. However, a long-term survey of flux dynamics is beyond the scope of the present paper, which concentrates on providing a "snapshot" of a particular period, as is commonly done in many field campaigns, and comparing branch-level fluxes with canopy-level fluxes measured in similar periods but different years. The reported data span from 14th of August to 1st September for branch level fluxes and from 17th of July to 8th of August for vertical profiles (42 days overall).

At page 10, lines 1-4, the following sentence has been added:

"The present study reports measurements for a particular period and compares them with canopy-level fluxes measured in similar periods but different years. A long-term survey of flux dynamics is beyond the scope of the present paper."

- Despite the concentration of several VOCs have been measured simultaneously by PTR-TOF-MS, the author reported in the manuscript only "the most important OVOC and volatile isorpenoids" (as written in line 1, page 16). The author should clearly define 'how' those oVOC and the isoprenoids have been selected/filtered form the multitude of protonated ions related to VOC (and/or fragment of VOC.) recorded by PTR-TOF-MS.

**The first reviewer also asked for this clarification. Please see the answer above.**

- I remind the author that 'concentrations' of VOC shown in Figure 1, may represent a larger area possibly including other sources/sinks than the mixed forest investigated by the author. Indeed only through a 'footprint analysis', a model that considers the concentration distribution of VOC with respect to wind speed and the turbulent air conditions occurring at the site, it can be sorted out the representativeness of the measured concentration on the surface area (Horst and Weil 1992, 1994; see also Schmid 1994 for details).

We agree with the reviewer that the concentrations shown in Figure 1 represent a larger area than the mixed forest in the immediate vicinity of the sample site. We carefully checked the manuscript and do not think it implies otherwise.

Footprint analysis is commonly used for canopy-scale surface-atmosphere fluxes measured using methods such as eddy covariance. In the case of flux measurements, a footprint is used to determine the surface area contributing to the vertical flux of scalars (such as species concentrations) at a specific height above the forest canopy. Footprint analysis is not used in the interpretation of species concentrations, because, even for short-lived gases, the upwind area contributing to the concentration at a particular location is likely to be quite large (10's to 100's of kilometers for a species with an atmospheric lifetime of a few hours.) For longerlived species (days to weeks), the "footprint" can be thousands of km. Furthermore, as the foregoing suggests, the concentration "footprint" will vary with the chemical lifetime of the species of interest. We note that the citations given by the reviewer are to the use of footprint analysis for flux measurements, e.g., Horst and Weil, 1992: "Footprint estimation for scalar flux measurements in the atmospheric surface layer."

In the current case, we are not reporting canopy-scale fluxes, but rather concentrations, so a footprint calculation would not be needed.

- The author discussed the results, i.e. of methanol fluxes (lines 9-24, page 30) by comparing VOC fluxes calculated in this manuscript from measurements collected at branch-level (and then scaledup at canopy-level after considering the average LAI and inhomogeneity of the canopy; as written in lines 20-24, page 16 and lines 1-4, page 17), with results found in the literature where VOC fluxes have been measured at canopy level by eddy covariance technique. The author should discuss and make the point on how the different approach in calculating the VOC fluxes may have produced different results (i.e. eddy covariance flux measurements provide emission/deposition fluxes from complex sources/sinks also including the soil, while branch measurements provide indications of VOC emissions mainly form leaves as the only source/sink).

We thank the reviewer for this suggestion. We added a discussion of the different approaches. At page 23, lines 8-14, the following sentence has been added:

"The scaling of measured branch-level fluxes to canopy-level assumes that the only contribution to the flux is from tree branches, while other possible sources and sinks, such as

**the soil or chemistry within the canopy, are neglected. This should be kept in mind when comparing the predicted canopy-scale fluxes to the measured eddy covariance fluxes. Some insight into the possible soil contribution to fluxes is provided by the concentration vertical profile data discussed below."**

- I am wondering whether the author can provide additional data on both the CO2 and water vapor exchange from branch enclosure measurements. Indeed, the measurements of CO2 assimilation rates would give information on the physiological performances of the leaves enclosed in the branch chamber; this would be particularly important to evaluate any damage eventually occurring to primary leaves metabolism following MVK absorption (in correlation with detoxification and thus MEK emission). In addition, measurements of water vapor exchange (even though at branch level) would provide basic information useful to both estimate the extent of stomata conductance and to partition the stomata and non-stomatal contribution to the uptake of (O)VOC (Fares et al. 2012) that has been mentioned several times throughout the manuscript.

We did not deploy specific detectors for CO2 and water vapor. The acquired PTR-ToF-MS data contain signals related to them, but there are several caveats. The reaction between the primary ion and CO2 does not proceed at collision rate and the water vapor signal is influenced by the primary ion signals. Since the signals for CO2 and water vapor were not explicitly calibrated for during the campaign, we prefer to avoid adding data of not assured quality. We agree with the reviewer that they would provide additional insights, but the findings of the paper do not rely on such data. Additional questions about the relationship between CO2, H2O and VOC exchange are beyond the scope of the present paper, but may be addressed in future studies.

- Since the 'compensation point' may vary on the basis of both physiological and physicochemical factors (as reviewed by Niinemets et al. 2014), when discussing this issue it would be better to indicate, at least, the environmental conditions under which this point has been determined. **We indicated the environmental conditions in the revised manuscript.**

At page 10, line 17-19, the following sentence has been added:

"During the ancillary experiments daytime temperature and PAR were in the ranges 27.1-28.2 °C and 894-1584 mmol m-2 s-1, respectively; while nighttime temperature was in the range 20.7-22.2 °C."

- Material and Methods section needs to be re-formatted and dramatically shorten it (now it is almost 10 pages long!). I suggest the author to either delete or move to 'Supplementary Information' most of the description of PTR-TOF-MS technical details (which is now 6 pages long) and simply mention citations of the many previous scientific works where PTR-TOF-MS has been already decribed. In fact, subsections 2.4.1 and 2.4.2 belong more to a textbook than to a scientific paper, while much of the 'Spectral analysis' subsection could be summarized in a table (i.e. by comparing VOC analysys by PTR-TOF-MS set in H3O+ and NO+ mode). Moreover, if not commercial, a schematic and/or a picture of the canopy-top branch enclosure employed for the measurements would be very much informative.

We agree with the reviewers that most of the description of PTR-TOF-MS technical details is more appropriate for the "Supplement". We therefore moved to the "Supplement" the subsection 2.4.1, 2.4.2 and 2.4.3 (now section S.4). The remaining section 2.4 has been shortened, referring the reader to the "Supplementary Information" for a detailed description. Section 2.4 now reads (page 10, lines 21-23, and page 11, line 1): "VOC measurements were performed by a PTR/SRI-ToF-MS 8000 (Ionicon Analytik GmbH, Innsbruck Austria) equipped with a switchable reagent ion system (Jordan et al., 2009), allowing either NO+ or H3O+ primary ion chemistry.

Details of the instrument operation are reported in the Supplementary Material."

**We also provided a picture of the branch enclosure system (Fig. S.1) and a table (Table S.1) summarizing the "Spectral analysis" subsections.**

MINOR ISSUES:

- Consistency I required throughout the manuscript! Please make use of either the term BVOC or VOC.

This point has been corrected. Only the term "VOC" is used in the revised manuscript.

- Is there any particular reason why fluxes are always expressed in "nmol m-2 hour-1", instead of "nmol m-2 second-1"? (in which all the measurements unit are indicated according to the International System of Units)

**We propose to keep "nmol m-2 hour-1" as it is commonly used and it is a convenient unit given the numbers involved.**

- Lines 12-13, page 1: I suggest to replace 'atmospheric reactor' with 'atmosphere'.

**This point has been corrected.**

- Line 14, page 6: the author should describe the plant species composition of the mixed forest that has been investigated; this can offer information to improve the discussion of sources/sinks of (O)VOC emission/deposition.

The first reviewer had a similar suggestion, please see the above answer.

- Line 21, page 6: it is written 'Influence', without specifying 'of what'?

Influences of anthropogenic VOC. At page 7, line 3, the sentence now reads: "Anthropogenic VOCs come from (...)".

- Lines 3-4, page 8: I wonder why the fonts of this sentence are in 'bold'...

Bold formatting has been removed.

- Lines 14-15, page 16: This is not clear, and I suggest the author to rephrase it.

The sentence has been rephrased as follows (page 11, lines 16-19):

"Figure 2 shows daytime and nighttime isoprene concentrations at different heights within and above the canopy. This vertical profile is consistent with an isoprene source within the canopy during the day, while it suggests that there is no isoprene emission nor deposition at night."

- Lines 13-14, page 17: I remind the author that also Brilli et al. (2016) showed MVK+MAC emission fluxes at canopy-level from a poplar plantation.

The suggested reference has been added (page 12, line 18).

- Line 4, page 18: the author should quickly explain 'why' 'the actual presence of emissions could not be proved.

The sentence has been modified as follows: "the actual presence of emissions could not be proved due to interferences from high levels of isoprene, as explained above."

- Lines 4-5, page 20: I suggest to remove both the words 'first' and to replace the last word with 'before'.

The sentence has been rephrased.

Page 14, lines 12-13, the sentence now reads: "(reduction of the carbonyl moiety versus reduction of the alkene moiety)".

- Lines 14-17, page 20: I remind the author that also Brilli et al. (2016) found emission and deposition fluxes of MEK through eddy covariance measurements at canopy level.

The following sentence has been added (page 16, lines 2-3):

"Brilli et al. (2016) reported positive and negative MEK fluxes at canopy level over a poplar plantation."

- Lines 19-20, page 22: the author should specify which kind of 'correlation' (i.e. linear?) We specified "linear correlation" (page 18, line 6)

- Lines 1-3, page 23: I remind the author that emission of alcohols (i.e. belonging to the mixture of Green Leaf Volatiles) does not depend to the plant species, but to the occurrence of specific situation (i.e. the presence of herbivores inducing mechanical damage to leaves).

At page 18, line 13, we removed ", however, GC-MS analysis of red oak volatiles did not report any potentially interfering alcohols (Helmig et al., 1999)"

- Line 16-17, page 24: the author meant 'benzaldehyde concentration', in air ?

Correct, we meant benzaldehyde concentration in the branch enclosure air.

At page 20, line 5, "in the branch enclosures" was added after "benzaldehyde concentration". - Lines 5-6, page 27: These citations are missing in the reference list.

The citations have been added to the list.

- Lines 7-8, page 28: I am wondering if the Model of Emissions of Gases and Aerosols from Nature has been never applied to simulate acetone emissions.

Yes, MEGAN has been applied to simulate acetone emissions (e.g. Messina et al, Atmos. Chem. Phys., 2016). We prefer not to mention it, as we were referring to the fact that the metabolic mechanisms leading to acetone release have not yet been proved and therefore, at present, no process-based model is capable of reliably describing biogenic acetone emissions. - Line 18, page 29: I remind the author that also Hüve et al. 2007 investigated the control of ethanol release by leaf expansion and stomata.

We thank the reviewer for this suggestion. The suggested reference has been added (page 25, lines 10 and 11).

We hope that the revised manuscript will be acceptable for publication in ACP. We thank you for your attention, and look forward to hearing from you.

Sincerely,

Luca Cappellin

**Field observations of Volatile Organic Compound (VOC) exchange in red oaks**

Luca Cappellin1,2, Alberto Algarra Alarcon2,4, Irina Herdlinger-Blatt3, Juaquin Sanchez1, Franco Biasioli2, Scot Martin1, Francesco Loreto5, Karena McKinney1

[revised manuscript text omitted]
     | $0.6 \pm 0.3$                              | $0.2\pm0.1$                                           | $0.4 \pm 0.2$                              | 2.2 *                                         | -1.9 *                                          |
| Acetaldehyde     | $50 \pm 30$                                | 11 ± 6                                                | $20\ \pm 10$                               | 170 *                                         | -20 *                                           |
| MEK              | $30 \pm 10$                                | $4\pm 3$                                              | $15 \pm 7$                                 | 140 *                                         | 0                                               |
| Acetone          | $90 \pm 50$                                | $20 \pm 30$                                           | $50 \pm 30$                                | 330 *                                         | 0                                               |
| MeOH             | $1000\pm800$                               | $300 \pm 100$                                         | $600 \pm 400$                              | 3000 *                                        | 0                                               |
| ISOP             | $0.2 \pm 0.1 \text{ x} 10^{54}$            | $\underline{200.0} \pm \underline{200.0 \times 10^5}$ | $0.10 \pm 0.0.7 \text{ x} 10^{54}$         | $1.0 \text{ x} 10^5 \text{ *}$                | <del>0.0 x1050</del>                 |
| Monoterpenes     | $20\pm10$                                  | $2 \pm 2$                                             | $10 \pm 7$                                 | 183 *                                         | 0                                               |
| MACR+MVK+ISOPOOH | -                                          | $0 \pm 2$                                             | -                                          | 7 n.s.                                        | -4 *                                            |

10

**Table 2.** Average measured emission and deposition rates at branch-scale in the upper canopy. August 25 - September 1. Data are reported as mean  $\pm$  standard deviation. Maximum emission and deposition rates are also reported as well as an indication of their statistical significance (\*: p<0.05; n.s.: not significant).

|                  | Average daytime
emission                | Average nighttime
emission                         | Average 24-hour
emission    | Maximum
emission rate
(1 h integration) | Maximum
deposition rate
(1 h integration) |
|------------------|--------------------------------------------|-------------------------------------------------------|--------------------------------|-----------------------------------------------|-------------------------------------------------|
|                  | [nmol_4m -2 _4h -1 ] | [nmol_/m -2 _/h -1 ]            | $[nmol_{-}/m^{-2}/h^{-1}]$     | [nmol_/m -2 _/h -1 ]    | [nmol_4m -2 _4h -1 ]      |
| Benzaldehyde     | -1 ± 2                                     | $0 \pm 1$                                             | -1 ± 2                         | 0.5 *                                         | -7 *                                            |
| Acetaldehyde     | $10 \pm 40$                                | -1 ± 10                                               | $3 \pm 18$                     | 100 *                                         | -40 *                                           |
| MEK              | $60 \pm 20$                                | $7\pm7$                                               | $30 \pm 20$                    | 460 *                                         | 0                                               |
| Acetone          | $50 \pm 20$                                | $10 \pm 10$                                           | $30 \pm 20$                    | 340 *                                         | 0                                               |
| MeOH             | $1000\pm1000$                              | $300\pm~200$                                          | $700\pm600$                    | 3000 *                                        | 0                                               |
| ISOP             | $0.4 \pm 0.4 \text{ x} 10^{54}$            | $\underline{200.0} \pm \underline{100.0 \times 10^5}$ | $0.2 \pm 0.2 \text{ x}10^{54}$ | $3.2 \text{ x} 10^5 \text{ *}$                | 0 <del>.0 x10</del> 5                |
| Monoterpenes     | $50\pm50$                                  | $20\ \pm 10$                                          | $40\ \pm 40$                   | 1370 *                                        | -50 n.s.                                        |
| MACR+MVK+ISOPOOH | -                                          | $-6 \pm 8$                                            | -                              | 0                                             | -23 *                                           |

5

**Table 3.** Summary of the ancillary experiments: Fumigation of forest red oak upper canopy branches with VOCs.

|         | Henry Law Constant
(water/air)
H cp at T O | VOC fumigation concentration range | Daytime
deposition | Nighttime
deposition |
|---------|------------------------------------------------------------------------|------------------------------------|-----------------------|-------------------------|
|         | $[mol \not-(m^{-3}-Pa^{-1})]$                                          | [ppbv]                             |                       |                         |
| MVK     | 2.6.10-1                                                               | 0 - 70                             | YES                   | YES                     |
| MACR    | 4.8.10-2                                                               | 0 - 50                             | YES                   | YES                     |
| ISOP    | $1.3 \cdot 10^{-4}$                                                    | 0 - 700                            | -                     | -                       |
| ACETONE | $2.7 \cdot 10^{-1}$                                                    | 0 - 70                             | -                     | -                       |
| MeOH    | 2.0                                                                    | 0 - 70                             | YES                   | YES                     |

**Figure captions**

10

25

Figure 1. Diel average measured mixing ratios and fluxes in enclosures containing upper canopy red oak branches at Harvard Forest. Mixing ratio of the inflow ambient air and of
the outflow air are shown in red and black, respectively. Diel average temperature and PAR during the measurement period are also shown. Error bars represent standard errors of the data points in each time interval.

**Figure 2.** Average daytime and nighttime vertical concentration profiles. Measured VOC concentrations at various heights within and above canopy are reported. Horizontal bars represent standard deviations.

Figure 3. Example of branch-level VOC flux measured as a function of MVK fumigation mixing ratio during daytime. The concomitant MVK uptake and emission of MVK reduction products (namely MEK, 3-buten-2-ol and 2-butanol) implies plant detoxification of MVK. Linear fit parameters are the following: MEK (R2=0.954; y=0.0249x+0.0869);
2-butanol (R2=0.928; y=0.0031x+0.0226); 3-buten-2-ol (R2=0.956; y=0.0102x-0.0102); MVK (R2=0.991; y=-0.0293x-0.0279).

**Figure 4.** Scatter plot of branch-level MEK fluxes versus MACR+MVK+ISOPOOH fluxes during darkness in the period 21-25 August. Linear fit parameters are the following:  $R^2=0.401$ ; y=-1.44x-0.24.

20 **Figure 5.** Example of benzaldehyde daily pattern for a day when benzaldehyde emission was detected. Measurements correspond to three different red oak upper canopy branches.

**Figure 6.** Flux measured as a function of mixing ratio for benzaldehyde during daytimes of the whole branch-level measurement campaign. Black triangles represent the period 25 August - 1 September, when deposition dominated, and grey dots correspond to the period 14-25 August, where mostly emission was measured. Linear fit parameters are the

following: R2=0.701; y=-75.59x+1.22.

**Figure 7.** Example of enclosure flux measured as a function of fumigation mixing ratio for methanol in the field.